# Paraplume: A fast and accurate antibody paratope prediction method provides insights into repertoire-scale binding dynamics

Gabriel Athènes[1,2], Adam Woolfe[2]*, Thierry Mora[1]*, Aleksandra M. Walczak[1]*

**1** Laboratoire de physique de l'École Normale Supérieure, CNRS, PSL University, Sorbonne Université, and Université de Paris, Paris, France, **2** Bio-Rad SAS, Marnes-la-Coquette, France

\* adam_woolfe@bio-rad.com (AW); thierry.mora@phys.ens.fr (TM); aleksandra.walczak@phys.ens.fr (AMW)

## Abstract

The specific region of an antibody responsible for binding to an antigen, known as the paratope, is essential for immune recognition. Accurate identification of this small yet critical region can accelerate the development of therapeutic antibodies. Determining paratope locations typically relies on modeling the antibody structure, which is computationally intensive and difficult to scale across large antibody repertoires. We introduce Paraplume, a sequence-based paratope prediction method that leverages embeddings from protein language models (PLMs), without the need for structural input and achieves superior performance across multiple benchmarks compared to current methods. In addition, reweighting PLM embeddings using Paraplume predictions yields more informative sequence representations, improving downstream tasks such as binder classification and epitope binning. Applied to large antibody repertoires, Paraplume reveals that antigen-specific somatic hypermutations are associated with larger paratopes, suggesting a potential mechanism for affinity enhancement. Our findings position PLM-based paratope prediction as a powerful, scalable alternative to structure-dependent approaches, opening new avenues for understanding antibody evolution.

## Author summary

Accurately identifying the small region of an antibody that binds the target antigen, the paratope, is important for immune recognition and designing effective therapies. Most existing approaches depend on 3D structural modeling, which is computationally demanding and limits large-scale analyses. We present a fast and scalable method that predicts paratopes directly from antibody sequences using protein language models. We show that asymmetric paratopes reflect biological binding mechanisms and correlate with the structures of cognate antigen epitopes.

**Data availability statement:** Paraplume is freely available for non-commercial use as a PyPI package and can be accessed at https://github.com/statbiophys/Paraplume/. The package is designed for ease of use and includes the complete pipeline, covering dataset preparation and paratope labeling, model training, and model inference, thereby enabling full reproducibility of our results. Both Paraplume and its variant Paraplume-S support GPU and CPU execution, as well as single-chain and paired- chain inputs. Users can readily retrain Paraplume on larger datasets with customized parameter settings, including selection of subsets among the six PLMs employed in this work. Details and data of all benchmark and application experiments are provided in https://zenodo.org/records/17021232 to ensure reproducibility.

**Funding:** The study was supported by European Research Council Proof of Concept 101185627 (AMW). We declare that this study received funding from Bio-Rad (AW). The funder collaborated directly in the study and was involved in the study design, analysis, and interpretation of data, the writing of this article, and the decision to submit it for publication.

**Competing interests:** I have read the journal's policy and the authors of this manuscript have the following competing interests: AW is a Bio-Rad employee and may hold shares and/or stock options in the company. We declare that this study received funding from bio-rad.

Applying our method to antibody repertoires, we find that affinity maturation in response to antigen exposure is associated with an increase in predicted paratope size. Our results open up new directions in exploring the functional consequences of antibody diversification and evolution.

## Introduction

Antibodies are specialized proteins of the immune system, produced by B cells, that recognize foreign pathogens, either neutralizing them directly or marking them for removal. This highly specific recognition is determined by the antibody's variable regions and is refined through a Darwinian process known as affinity maturation, which B cells undergo after encountering an antigen. During this process, the genes encoding the variable regions undergo somatic hypermutation, and B cells producing higher-affinity antibodies are selectively expanded. The paratope comprises specific amino acids in the variable regions of the antibody that directly interact with residues on the target antigen, known as epitopes, upon binding (Fig 1A). This interaction determines the antibody's binding specificity and affinity, both of which are essential for an effective immune response. Mapping the specific location of the paratope has important applications in biotechnology and medicine, especially in the design of therapeutic antibodies, as accurate predictions of antibody binding sites can help identify key residues for targeted mutations that modify binding properties [1] or that should be avoided during engineering of antibodies for enhanced developability.

However, experimental methods for determining antibody-antigen binding interactions are slow and resource-intensive [2]. In contrast, computational methods such as molecular docking have been developed as a more efficient alternative, offering faster, lower-cost approaches to predict how antibodies and antigens can bind [3]. While promising, these tools still face limitations in accuracy [4], especially at a scale required for high-throughput applications [5], and they require the 3D structures of both antibodies and antigens. Although the recent release of Alphafold 3 [6] shows an improvement in modelling accuracy of the antibody-antigen complex, it is limited in the antibody-docking task [7] and requires the antigen sequence.

To address these challenges, numerous methods have been developed for predicting antibody paratopes. Parapred [8], a freely accessible sequence-based tool, uses convolutional neural networks to extract local sequence features and recurrent neural networks to capture long-range dependencies. While practical, Parapred is limited to predicting paratopes within the complementarity-determining regions (CDRs) of the antibody, requiring sequence numbering as a prerequisite. These 6 CDRs (three in the heavy chain and three in the light chain) encompass the majority of the paratope, thereby simplifying the training of supervised models. However, recent advancements in methods that leverage antibody 3D structural information have surpassed Parapred in performance, leading state-of-the-art paratope prediction approaches to predominantly rely on either experimentally determined structures or high-quality modeled counterparts. Paragraph [9] models the 3D antibody structure using AbodyBuilder [10] and Ablooper [11], represents the structure as

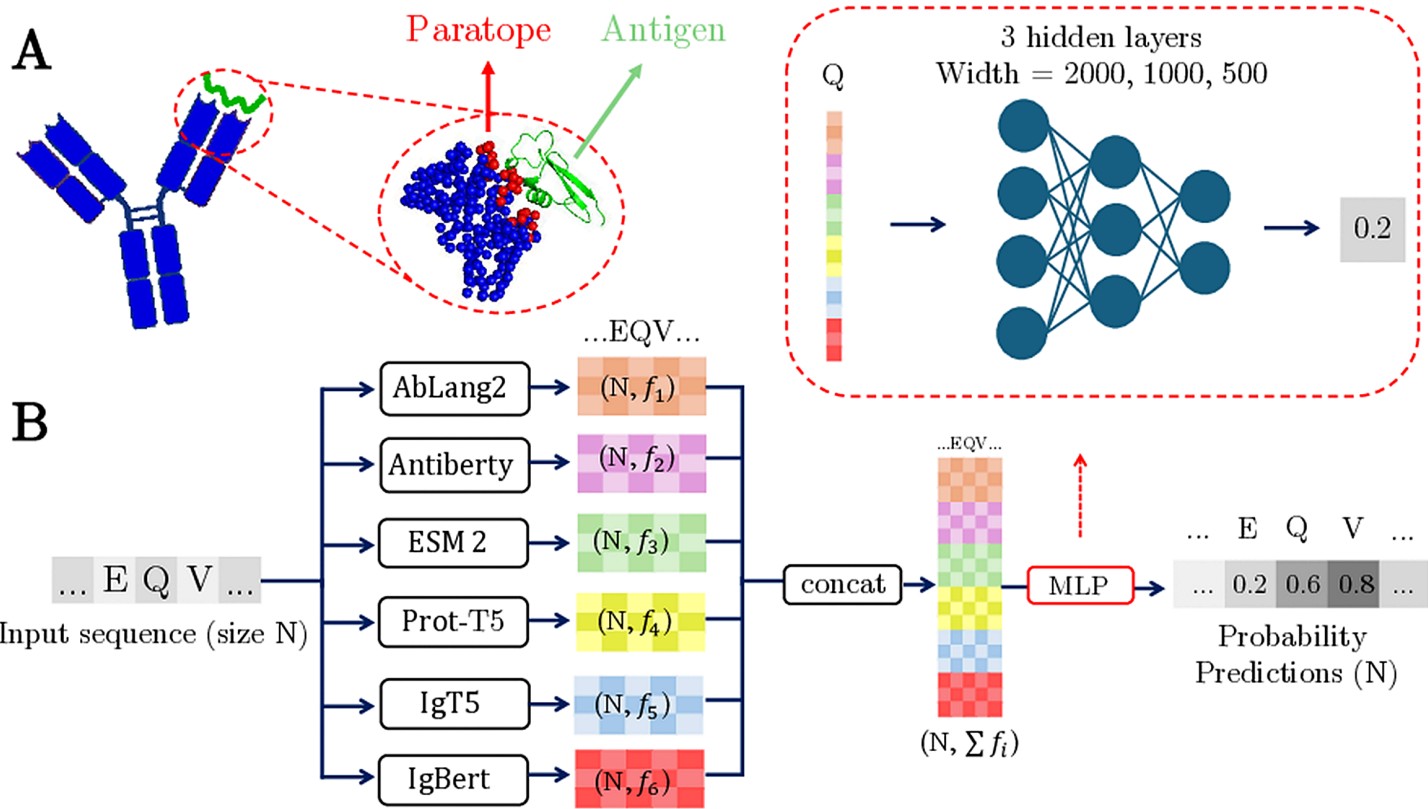

Fig 1. **(A) Antibody (blue) binding to an antigen (green), illustrated using the structure of the variable domain ($F_v$ region) of the mouse anti-lysozyme antibody (PDB Code: 1BVK).** Amino acids are represented using carbon alpha $C_\alpha$ atoms, and the paratope is colored red. Amino acids are labeled as belonging to the paratope if any non-hydrogen atom is within a distance of 4.5 Å of a non-hydrogen antigen atom. (B) The pipeline used for paratope prediction. The antibody sequence is given as input to protein language models (PLMs), the last embedding layer of which is concatenated and fed to a multi objective multilayer perceptron (MLP). The MLP calculates probabilities of amino acids belonging to a paratope.

a graph based on amino acid distances, and then uses equivariant graph neural networks [12] for the paratope prediction task. Similarly, methods like PECAN [13] and MIPE [14] require the 3D structures of both the antibody and antigen to predict the paratope. However, experimentally determined 3D structures are not always readily available, and generating accurate 3D models introduces additional challenges. This reliance on structure prediction models not only leads to a significant drop in performance, as observed in [14], but also requires time-intensive pre-processing steps to compute interacting residues. These limitations underscore the need for more precise and scalable computational models that can effectively identify antibody binding sites.

Powered by the Transformer architecture, protein language models (PLMs) pretrained on huge databases of protein sequences have been applied to tasks such as secondary structure prediction and contact map estimation [15,16]. Their ability to extract structural and functional information from sequence data alone makes them especially valuable for predicting antibody binding sites in the absence of structural information. We introduce Paraplume, a sequence-based, antigen-agnostic paratope inference method that overcomes data limitations by leveraging embeddings from six PLMs and achieves state-of-the-art performance on three independent datasets.

Our work introduces several conceptually new elements that go beyond incremental benchmarking improvements: (i) a multi-PLM ensemble architecture that improves both scalability and accuracy, supports inference over the entire variable region, and provides flexibility to process either single or paired chains, (ii) an asymmetry-based upper-bound

analysis defines a theoretical limit for sequence-only paratope inference on the benchmark datasets used in this study, (iii) paratope-weighted sequence embeddings that explicitly emphasize functionally relevant residues and can be used for downstream predictive tasks, and (iv) the first repertoire-scale paratope analyses enabled by the method's computational efficiency, and previously inaccessible due to the computational constraints of prior methods. Using Paraplume, we compared naive and antigen-exposed antibody repertoires and identified a clear signal of paratope evolution.

## Results

### Paraplume

Paratope prediction consists of assigning a label 1 to an amino acid if it belongs to the paratope and 0 otherwise. Supervised methods construct training and testing datasets by annotating amino acids with paratope labels using the experimentally determined 3D structures of antibody-antigen complexes available in SabDab [17]. Concretely, an antibody amino acid is labeled 1 if at least one of its non-hydrogen atoms is within 4.5 Å of a non-hydrogen atom of the antigen.

The main challenge in using structural data to train supervised models for paratope inference is the limited availability of structures in SAbDab. To mitigate this issue, we leverage information from millions of sequences by representing all amino acids from the variable region as embeddings derived from protein language models (PLMs). PLMs are typically trained in an unsupervised manner on large protein sequence datasets. Antibody-specific PLMs are either trained directly on large collections of antibody sequences or adapted from general protein PLMs through finetuning. These models produce embedding vectors that contain information not just about the amino acid itself, but also about its sequence context through the attention mechanism. While most approaches using PLMs rely on a single model, we hypothesized that concatenating embeddings from multiple PLMs could provide complementary information not captured by any individual model alone. Specifically, each amino acid is represented as the concatenation of embeddings from six language models: AbLang2 [18], Antiberty [19], ESM-2 [15], IgT5 [20], IgBert [20], and ProtTrans [16] (cf. Methods). These concatenated embeddings are then input to a Multi-Layer Perceptron (MLP) with three hidden layers of size 2000, 1000 and 500, that uses paratope labels for training (Fig 1B). Once the embeddings are computed, training the model becomes computationally feasible using only a CPU.

Here, we introduce Paraplume, the resulting sequence-based supervised paratope prediction model (Fig 1B). Paraplume assigns a probability to each amino acid in the input sequence, reflecting its likelihood of being part of the paratope. It is trained by minimizing the Binary Cross-Entropy loss between the predicted probabilities and the true labels (cf. Methods). Details of the model parameters and training protocol are described in Methods. Although the three-dimensional structure is essential for generating the labels used during training, Paraplume does not require structural information to make predictions. Paraplume takes as input either the heavy chain, the light chain, or paired heavy and light chains, and makes predictions solely based on sequence data (cf. Methods). Paraplume is also antigen-agnostic, meaning it does not require any antigen-specific information for its predictions. A key advantage of Paraplume's sequence-based design is its computational efficiency, allowing paratope predictions for 1000 sequences in 3 minutes (50 seconds if only using one ESM embedding) using a single GPU (cf. S1 Fig), facilitating large-scale analysis of antibody sequence repertoires.

### Embedding contribution and interpretability

To test the hypothesis of PLM embedding complementarity, we evaluated the individual contributions of six protein and antibody language models (cf. Methods). Concatenating embeddings consistently improved benchmark accuracy, indicating that each PLM captures distinct and complementary aspects of antibody sequence information. Importantly, these gains were not simply due to increased model capacity: expanding the MLP input layer by stacking multiple copies

of the same embedding (ESM) resulted in substantially worse performance compared to combining different PLMs (S2 Fig).

We conducted a control experiment with Prost-T5 [22], a structure-aware PLM that integrates 3D information during pre-training while requiring only sequence input at inference. As shown in S2 Fig, Paraplume maintains stronger performance without this model, indicating that it can capture structural features from PLMs without the need for explicitly structure-aware models. While integrating Prost-T5 with our PLMs could be a promising direction for future work, we chose to exclude it from our model set because its training relied on AlphaFold-predicted structures. Since some of the structures used by AlphaFold overlap with complexes in our benchmark test sets, this raised concerns about potential data contamination.

Because larger embedding sets increase inference time and memory usage, we report parameter counts, RAM requirements, and inference speed for 100 sequences across all PLM configurations (S2 Fig). Leave-one-embedding-out experiments (S1 Table) showed that the full embeddings generally ranked first or second in more cases. Additionally, we used PCA to reduce embeddings across all six PLMs (S1 Table) to see if smaller vectors could maintain performance. The results revealed a strong decline. However, several reduced configurations achieved competitive accuracy at lower computational cost (S2 Fig), motivating a lighter alternative: Paraplume-S, a fast model using only ESM embeddings. A comparative analysis of inference-time compute requirements and estimated $CO_2$ emissions for Paraplume, Paraplume-S, and Paragraph under CPU and GPU conditions is shown in S1 Fig. We additionally provide tutorials and command lines examples that enable straightforward retraining of Paraplume on any combination of PLMs with a custom dataset, as well as the use of these customized models for inference.

Finally, the simplicity of the MLP architecture limits interpretability. To gain deeper insight into the contributions of individual PLMs to Paraplume's predictions, we computed Shapley values [21] as a model-agnostic measure of feature importance (cf. Methods). This approach yields normalized PLM-level importance weights that sum to one, providing a concise and interpretable quantification of each model's contribution. Paraplume enables visualization of importance profiles either at the sequence level or aggregated across datasets (S3 and S4 Figs), together with sequence-level predictions across all amino acids. Using this approach, we analyzed averaged predictions and PLM importance scores across IMGT positions in the Paragraph test set, with Paraplume trained on the Paragraph training set (Fig 2). Specifically, for each IMGT position, prediction values and PLM importance scores were averaged over all sequences in which that position is present. Importantly, while Paraplume primarily predicts paratopes within the three CDRs, it uniquely captures the recently characterized DE loop located in the framework region, which is inaccessible to methods that restrict predictions to the CDR$\pm$2 residues like Paragraph or Parapred (Fig 2) [23]. ESM, despite being the strongest overall performer, shows a marked reduction in relative importance within the four loops, which represent the most variable and structurally unpredictable regions of the heavy chain. In contrast, antibody-specific PLMs such as Ablang2 and Antiberty display increased importance in this region. This difference reflects their training objectives: ESM is a general protein language model not fine-tuned on antibody sequences, whereas Ablang and Antiberty are trained exclusively on antibody repertoires. Together, they highlight the complementarity between general-purpose and antibody-specialized PLMs.

## Performance comparison

We evaluate the performance of Paraplume in comparison to existing paratope prediction methods across three datasets using four evaluation metrics. The PECAN dataset comprises 460 antibody-antigen complexes with paired heavy and light chains, all resolved at sub-3 Å resolution, and is divided into 205 training, 103 validation, and 152 test samples. The Paragraph dataset, extracted from the Structural Antibody Database (SAbDab) as of March 31, 2022, consists of 1,086 antibody-antigen complexes, partitioned into training, validation, and test sets in a 60-20-20% split. The MIPE dataset includes 626 antibody-antigen complexes, with 90% allocated for training and 10% for testing. Sequence redundancy and

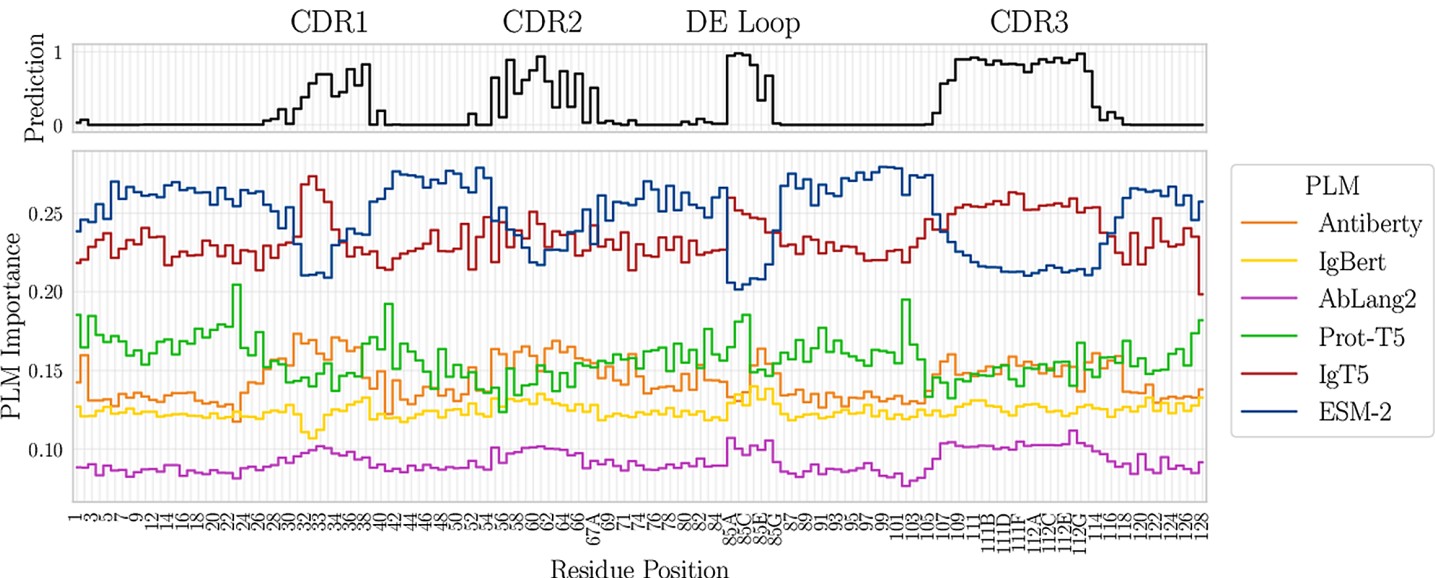

**Fig 2**. **Visualization of the average Paraplume prediction probability (top) and PLM raw importance score (bottom) across IMGT positions for 218 unseen sequences.** IMGT positions are labeled every two residues for clarity. Predictions are made on the Paragraph test set using Paraplume trained on the Paragraph training set. The importance scores were computed using Shapley values, as described in Methods [21]. Paratope predictions localize primarily to CDRs, where general PLMs have decreased contribution over antibody-specific PLMs. Notably, Paraplume identifies binding residues within the non-canonical FW3 DE loop, a region inaccessible to CDR-restricted methods.

train-test leakage were already addressed by the authors who curated the MIPE, Paragraph, and PECAN datasets. For completeness, we report the sequence similarity between the train and test splits of each dataset in S5A Fig.

Paraplume is underlied by several choices of architecture and hyperparameters detailed in Methods. The results presented below were obtained for the best performing model. All benchmark evaluations are done with identical hyperparameters and modeling choices, without tuning on individual datasets. Model performance is assessed using four metrics: the precision-recall area under the curve (PR AUC) and the receiver operating characteristic area under the curve (ROC AUC), which evaluate classification performance in imbalanced datasets; the F1-score (F1), representing the harmonic mean of precision and recall; and the Matthews correlation coefficient (MCC). Both F1 and MCC are computed using the standard 0.5 threshold to binarize predictions. Following the approach used for other methods, each metric was averaged over all proteins in the test set.

The benchmarked methods vary significantly in their approaches: some predict paratopes directly from sequence data (Parapred, Paraplume), others rely on modeling the 3D structure from the sequence (Paragraph, PECAN, MIPE)—a preprocessing step that reduces scalability—and some make predictions based on the experimentally determined structure of the antibody alone or in combination with the antigen (Parasurf-Fv, and versions of Paragraph, PECAN and MIPE). Because the experimentally determined antibody structures used for training and testing by these methods are derived from antibody-antigen complexes, each structure serves both as model input and for paratope labeling. Given that antigen binding can induce substantial conformational changes in antibodies [24], this raises concerns about the generalizability of such models to unbound antibody structures. To ensure a fair comparison, in Table 1 we compare Paraplume with methods that do not take experimentally determined structures as input: Parapred, PECAN, Paragraph, MIPE, and the baseline method described in [9]. Paragraph and PECAN use ABodyBuilder [10] for structure modeling from the sequence, while MIPE uses AlphaFold2 [25]. In contrast, Parapred and Paraplume do not require structure modeling. Additionally, MIPE and PECAN incorporate antigen information in their predictions. For the PECAN and Paragraph

Table 1. **Comparison of methods that use sequences as inputs.** Paragraph and PECAN model the 3D structures from the sequences with ABody-Builder (ABB) [10], while MIPE uses AlphaFold2 (AF2), and requires both antibody and antigen sequences. All other methods operate directly on sequences without requiring structural modeling. Performance metrics (PR AUC, ROC AUC, F1 score, and MCC) with additional model characteristics (structure modeling free and antigen agnostic) for models evaluated on PECAN, PARAGRAPH, and MIPE datasets. The highest value in each column is in bold, the second best is underlined.

**Using sequences as inputs**

| Model | PECAN Dataset | | | | Paragraph Dataset | | | | Mipe Dataset | | | | Structure modeling free | Antigen agnostic |
|---|---|---|---|---|---|---|---|---|---|---|---|---|---|---|
| | PR | ROC | F1 | MCC | PR | ROC | F1 | MCC | PR | ROC | F1 | MCC | | |
| Baseline | 0.626 | <u>0.952</u> | 0.665 | 0.635 | 0.624 | <u>0.952</u> | 0.622 | 0.654 | 0.465 | 0.931 | 0.536 | 0.177 | ✓ | ✓ |
| Parapred | 0.646 | 0.930 | - | - | - | - | - | - | 0.652 | 0.868 | - | 0.503 | ✓ | ✓ |
| Paragraph (ABB) | 0.696 | 0.934 | **0.685** | <u>0.654</u> | <u>0.725</u> | 0.934 | <u>0.696</u> | <u>0.669</u> | 0.689 | <u>0.937</u> | <u>0.617</u> | 0.596 | ✗ | ✓ |
| PECAN (ABB) | 0.675 | <u>0.952</u> | - | - | - | - | - | - | - | - | - | - | ✗ | ✗ |
| MIPE (AF2) | - | - | - | - | - | - | - | - | **0.723** | 0.910 | <u>0.617</u> | 0.531 | ✗ | ✗ |
| Paraplume | **0.730** | **0.963** | <u>0.682</u> | **0.657** | **0.758** | **0.966** | **0.701** | **0.676** | <u>0.716</u> | **0.962** | **0.651** | **0.632** | ✓ | ✓ |

datasets, performance metrics for all methods were obtained from [9]. For the MIPE dataset, results were taken from [14], with the exception of Paragraph. To present Paragraph in the most favorable light, we retrained the model and evaluated its performance using inputs generated from structures predicted by ABodyBuilder3 (ABB3) [26]. We selected ABB3 because it is the latest iteration of ABodyBuilder, providing state-of-the-art accuracy for antibody structure prediction while remaining fully consistent with the procedure used in the original Paragraph study. Note that this method may slightly overestimate Paragraph's performance, as some sequences in the MIPE test set are also in the ABB3 training data. However our method being sequence based, no data leakage can occur for Paraplume between the structure-prediction training set and the benchmark test sets. As with other methods, Paraplume was trained and evaluated separately on each of the three datasets using their respective predefined splits. For the PECAN and Paragraph datasets, Paraplume was trained using 16 random seeds. For each seed and dataset, early stopping was applied by retaining the model weights that achieved the highest PR AUC on the validation set, and performance was then evaluated on the corresponding test set. On the MIPE dataset, we performed a 5-fold cross-validation on the training-validation set, consistent with other methods [14] and [27]. For each fold we trained our model on the training set, retained the weights that maximized the PR-AUC on the validation set for testing on the independent test set. The reported results are averaged over the 5 folds and 5 seeds as done in [14]. Using only antibody sequence information, Paraplume outperformed all other methods across all four evaluation metrics on the Paragraph datasets, and for three out of four metrics for the PECAN and MIPE datasets (Table 1).

We show an example of Paraplume's predictions, trained on Paragraph's train set, compared to the ground truth labels of an antibody specific to the aTSR domain of a circumsporozoite protein (PDB: 6B0S) from Paragraph's test set (S6A Fig). Paraplume correctly identified all 23 experimentally determined paratope residues (TPR = 100%) while falsely labeling 9 of 205 non-paratope residues (FPR ∼ 4.4%). Paraplume successfully predicts paratope residues located in the framework region (S6B Fig), which is not achievable with methods limited to predictions within the CDR ±2 region such as Paragraph or Parapred. Finally, since paired chain data is often unavailable in large-scale bulk sequencing studies, it is important to assess whether Paraplume maintains reliable performance on single-chain sequences. To this end, we evaluated Paraplume on single-chain variants (cf. Methods for details) and observed only a minor decrease in performance, supporting its applicability to heavy chain only repertoires (S2 Table).

## Combining paraplume and paragraph for experimentally determined structures

Recent studies [9,14] have demonstrated notable differences in performance between models using experimentally determined structures and those relying on predicted structural models. Among methods that use experimentally determined structures (Table 2), some such as Paragraph, show improved performance within the CDR regions compared

**Table 2. Comparison of methods that use experimentally determined structures as inputs.** Performance metrics (PR AUC, ROC AUC, F1 score, and MCC) with additional model characteristics (antigen agnostic) for models evaluated on PECAN, PARAGRAPH, and MIPE datasets. The highest value in each column is in bold, the second best is underlined.

**Using experimentally determined structures as inputs**

| Model | PECAN Dataset | | | | Paragraph Dataset | | | | Mipe Dataset | | | | Antigen agnostic |
|---|---|---|---|---|---|---|---|---|---|---|---|---|---|
| | PR | ROC | F1 | MCC | PR | ROC | F1 | MCC | PR | ROC | F1 | MCC | |
| Paragraph | <u>0.754</u> | 0.940 | **0.703** | 0.674 | 0.770 | 0.939 | **0.719** | **0.692** | 0.742 | 0.943 | 0.651 | 0.634 | ✓ |
| Parasurf-Fv | 0.733 | <u>0.955</u> | 0.647 | 0.612 | **0.793** | <u>0.967</u> | 0.698 | 0.676 | **0.781** | **0.967** | **0.690** | **0.659** | ✓ |
| PECAN | 0.700 | <u>0.955</u> | - | - | - | - | - | - | 0.713 | 0.915 | - | 0.558 | ✗ |
| MIPE | - | - | - | - | - | - | - | - | 0.741 | 0.927 | 0.627 | 0.554 | ✗ |
| Paraplume-G | **0.772** | **0.965** | <u>0.697</u> | **0.675** | <u>0.791</u> | **0.968** | <u>0.704</u> | <u>0.683</u> | <u>0.753</u> | <u>0.964</u> | <u>0.663</u> | <u>0.648</u> | ✓ |

to Paraplume, but this advantage is lost in the framework regions, or when using modeled structures instead of experimentally determined structures (S3 Table). This could be because Paragraph is trained in the CDR$\pm$2 region, where the paratope-to-non-paratope ratio is higher (1 : 3 compared to 1 : 10 in the whole variable region), thereby stabilizing training.

To further increase performance across the entire sequence we developed Paraplume-G (Graph-based Paraplume), which uses structural information and combines the strengths of both Paragraph and Paraplume. Specifically, Paragraph, trained using the parameters described in the original paper, is used to predict residues in the CDR$\pm$2 region, while Paraplume is applied to predict residues outside this region. Concretely, the input structure is processed by Paragraph, which outputs residue probabilities only for the CDR$\pm$2 region. In parallel, the input antibody sequence derived from the structure is processed by Paraplume to generate predictions for all positions. We retain Paragraph's probabilities for the CDR$\pm$2 residues and substitute these into Paraplume's output, using Paraplume's predictions only for the remaining framework positions. This yields a single, contiguous set of residue probabilities spanning the entire sequence.

Table 2 presents results for methods that rely on experimentally determined structures, comparing Paraplume-G with Paragraph, Parasurf [27], PECAN, MIPE, and the baseline method described in [9]. Performance metrics for Parasurf and MIPE across the three datasets were obtained from their respective publications. For PECAN, results were taken from [9] for the PECAN dataset and from [14] for the MIPE dataset. For Paragraph, we used the results on the PECAN and Paragraph datasets from [9] and retrained Paragraph using experimentally determined structures for the MIPE dataset, following the approach described in [9], averaging the results across 16 different seeds. We observed significantly higher performance on the MIPE dataset compared to the results reported in [14]. Paraplume-G demonstrated performance comparable to state-of-the-art methods for experimentally determined structure-based paratope prediction, ranking first or second across all 12 metrics derived from the three datasets. It outperformed Parasurf on 7 of the 12 comparison points and surpassed Paragraph on 9 of them.

## Calculating performance upper bounds of paratope prediction using identical antibody arms in antibody-antigen structures

Proteins are not rigid structures but instead exist as ensembles of conformations that fluctuate over time. A recent study [28] suggests that a single antibody can adopt multiple conformations, underscoring the structural flexibility of CDR loops in the context of antigen recognition. This suggests that paratope definition may not be a straightforward problem, setting a potential upper bound of any paratope prediction method. To explore the extent to which paratope definition may vary, we curated a dataset of 470 antibody-antigen complexes from the SabDab database in which a single antibody binds two identical antigens, one for each arm, allowing direct comparison of ground-truth paratope annotations across

the two arms (Fig 3A, see Methods for details). We quantified the variability between the two antibody chains using a metric we define as paratope asymmetry (and analogously epitope asymmetry), which counts the number of residues present in the paratope or epitope in one arm, but not the other (see Methods and S7A and S7E Fig). We found that paratope and epitopes can vary by more than 10 amino acids between arms (S7B and S7F Fig). To account for size-dependent effects (S7C and S7G Fig), we also define a normalized version of these metrics based on the total paratope or epitope size (cf. Methods) and S7D and S7H Fig).

We investigated potential sources of asymmetry, such as antigen type, the structure determination method, and PDB resolution, but found only weak correlations (S8 Fig). By contrast, we observed a strong correlation between normalized paratope and epitope asymmetry (Fig 3C), indicating that structural changes in the antibody are closely mirrored by changes in the antigen interface, as illustrated in Fig 3B. This suggests that the observed asymmetry reflects real biological dynamics rather than technical artifacts. To assess the robustness of this relationship, we performed a sensitivity analysis using a dataset in which heavy-chain, light-chain, and antigen sequence differences were not fixed at zero but were instead required to be below 20 amino acids. We examined how normalized paratope asymmetry varied with differences in arm and antigen sequence (S9A and S9B Fig) and additionally compared normalized paratope versus epitope asymmetry for two groups of sequences: those that differed by 1 to 10 amino acids and 10 to 20 amino acids (S9C and S9D Fig). Finally, we report these relationships for amino acid differences of 1, 2, 3, and 4 in S9 FigE–H).

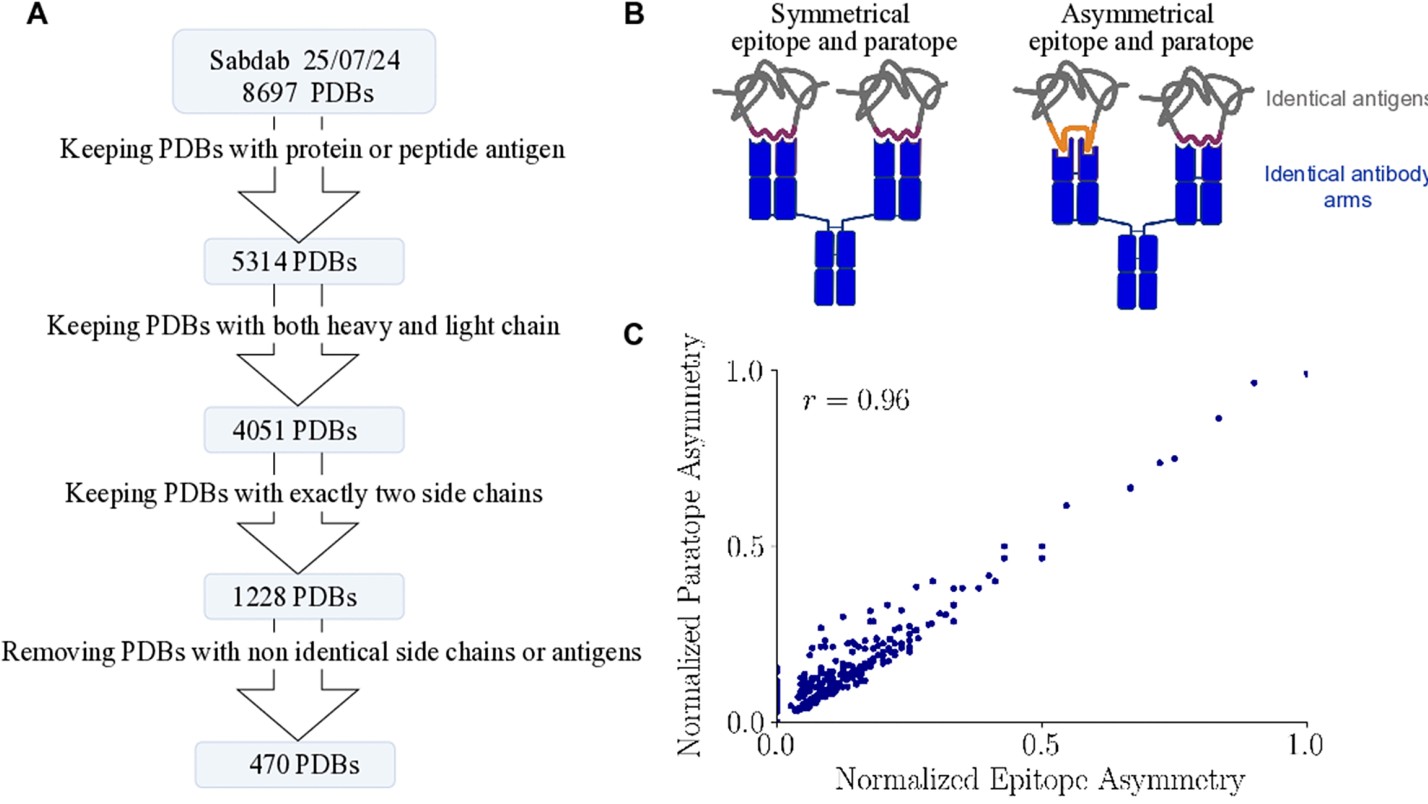

**Fig 3. (A) Dataset curation for the epitope asymmetry analysis, with the number of PDB structures at each stage.** (B) Cartoon of antibody-antigen complexes with symmetric and asymmetric paratopes and epitopes. An antibody side chain paratope binds to an epitope on the antigen side chain. In the asymmetric case two identical antibody sequences bind different epitopes, using different paratopes. (C) Normalized paratope asymmetry correlates strongly with the normalized epitope asymmetry (Pearson correlation coefficient), where each point represents a distinct antibody-antigen complex.

**Table 3**. F1 score and MCC for paratope prediction for the *Upper Bound* and *Paraplume* conditions across the PECAN, Paragraph, and MIPE datasets.

| Method | PECAN | | Paragraph | | Mipe | |
|---|---|---|---|---|---|---|
| | F1 | MCC | F1 | MCC | F1 | MCC |
| Upper Bound | 0.956 | 0.953 | 0.955 | 0.951 | 0.955 | 0.952 |
| Paraplume | 0.674 | 0.647 | 0.709 | 0.683 | 0.689 | 0.663 |

This biological variability provides an empirical upper limit on the performance of sequence-based paratope prediction models. For each of the three benchmark test datasets, we extracted the subset of structures present in our curated, fully filtered set (470 sequences) comprising 67 Paragraph sequences, 13 MIPE sequences, and 51 Pecan sequences. We measured the F1 score by treating one arm's labels as the "ground truth" and the other as "predictions". Across all three datasets, we found this upper bound to be consistently around 95% for the F1 score (Table 3, with our model's performance included for comparison). Importantly, this value should not be interpreted as an absolute performance ceiling: it depends on the specific distance cutoff used for defining paratopes as well as the composition of the dataset. To better illustrate the variability across antibody-antigen complexes, we additionally report the full distribution of arm-to-arm F1 and MCC scores on the 67 sequences from Paragraph test set (S10 Fig). For comparison and because antibodies in the benchmark test datasets may show differences between their two arms (for example due to missing residues), we also report in S10 Fig the distributions of F1 and MCC scores for 13 additional antibodies from the Paragraph dataset in which the two arms are not identical. The upperbound computed including these antibodies remains essentially unchanged (F1 Score: 0.953, MCC: 0.949). To assess how this variability affects our model, we further analyzed its predictions on ambiguous residues, defined as those with discordant paratope labels between the two arms, and on consistent residues, where labels agreed. We observed that predictions for ambiguous residues were more frequently distributed around 0.5, indicating reduced confidence and greater difficulty in predicting paratopes for residues subject to biological variability (S11 Fig). Together these observations highlight a critical limitation: even under ideal conditions, where each antibody binds to a single antigen type, perfect prediction remains impossible due to natural structural variability. In fact, as antibodies may interact with a diverse range of antigen types, we expect the maximum achievable performance to be even lower. To set the corresponding upper bound, one would need to compare 3D structures of the same antibody binding to distinct antigens, and measure the difference in their paratopes. However no such data is available to our knowledge. Thus, how much room there is left for the improvement of computational paratope prediction methods remains an open question.

## Application to large scale antibody sequence datasets

**Probability of amino acid belonging to a paratope correlates with impact on binding affinity.** To demonstrate the model's applicability in exploring antibody-antigen binding, we analyzed a dataset from Phillips et al. [29], comprising antibody sequences with experimentally measured binding affinities to three influenza strains (H1, H3 and FluB). This dataset was constructed by introducing all possible combinations of 16 mutations that differentiate the broadly neutralizing antibody (bnAb) CR9114 from its germline, totaling $2^{16}$ unique sequences. The study revealed that broad neutralization emerges sequentially, with binding initially increasing for the H1 strain, followed by H3, and finally FluB, as mutations accumulate in the germline. Moreover, the mutational effects on binding affinity exhibit a nested structure, where antibodies binding to FluB also bind to H3, and those binding to H3 also bind to H1. To examine the role of the paratope for binding affinity, we used Paraplume, trained on the complete expanded dataset from [9], excluding the 2 PDB structures of the CR9114 bnAb (PDB labels 4FQI and 4FQY), to predict the paratopes of all $2^{16}$ antibody variants. For each strain, we excluded antibody sequences that did not exhibit measurable binding affinity to the corresponding antigen ($-\log(K_d) = 7$ for H1 and $-\log(K_d) = 6$ for H3 and FluB), resulting in separate subsets of binders for each strain. Within

each subset, and for each of the 16 specific mutations, we identified sequence pairs that differed only by that particular mutation. For each pair, we computed the absolute difference in the predicted probability of the mutation site belonging to a paratope and the absolute difference in binding affinity for the strain. For each strain subset we then averaged these differences across all pairs to obtain the mean absolute difference in probability of the mutation belonging to a paratope, $\overline{|\Delta \text{Paratope Probability}|}$, and the mean absolute difference in binding affinity, $\overline{|\Delta \log K_D|}$. As a result we obtain the average change in the probability that this residue is part of the paratope for each of the 16 mutations, which correlates positively with the average change in the binding affinity, for each strain (Fig 4A). Mutations that result in significant changes in the probability of the amino acid to belong to the paratope suggest that these mutations are likely to influence the binding of the amino acid at that position, thereby affecting affinity.

**Mutations increase paratope size.** We next investigated the impact of mutations on paratope size, approximated as the sum of the probabilities of belonging to a paratope for all amino acids in both the heavy and light chains. To validate this metric as a proxy for physical size, we compared Paraplume scores against the ground-truth buried solvent-accessible surface area (bSASA) using the Paragraph test set, with Paraplume trained on the Paragraph training set. We observed that our proxy correlated more strongly with the ground truth (Pearson $r = 0.392$, $p = 1.4 \times 10^{-6}$) than bSASA values derived from AlphaFold-Multimer structure predictions (Pearson $r = 0.205$, $p = 1.4 \times 10^{-2}$), despite the latter using antigen sequence data (cf. S12 Fig). While paratope size could also be computed by thresholding predicted probabilities and counting residues above the cutoff, we observed that binarizing the probabilities at various thresholds reduced the correlation with ground-truth bSASA compared to summing all probabilities. This supports our choice of using the summed probability as the paratope size proxy (cf. S13 Fig).

Analysis across all antigens reveals a positive correlation between paratope size and mutation count (Fig 4B). This correlation is absent in non-binding antibodies (Fig 4B, bottom panel), implying that those unable to bind strongly to any of the three strains likely failed to develop a corresponding paratope. However, the interpretation of a computationally identified paratope for a non-binding antibody is unclear. Since the model was trained on antibodies with a defined antibody-antigen complex, it might be biased, resulting in overestimated paratope predictions for antibodies that do not interact with an antigen.

**Validation on antibody repertoires.** While the analysis of all intermediates between a naive and a matured antibody allows us to get a full picture of the sequence landscape for that particular pair, these sequences are not representative of actually explored variants in naturally occurring lineages found in antibody repertoires. To address this limitation, we analyzed data from Gerard et al. [30] consisting of IgG paired heavy and light chain sequences from two mice immunized with tetanus toxoid (TT). Antigen-binding, IgG-expressing B cells were isolated using a fluorescence-based droplet assay within a microfluidic sorting system, yielding 1,390 VH/VL pairs with ~93% of them binding to the tetanus toxoid (TT) antigen. This resulted in a TT-immunized repertoire of TT-specific antibodies, which we compared to a naive antibody repertoire from mice of the same species reported by Goldstein et al. [31]. The naive repertoire was subsampled to match the heavy chain V gene family distribution observed in the TT-immunized repertoire. For each antibody repertoire, we inferred germline sequences, quantified hypermutation (SHM) counts, performed paratope predictions, and identified clonal lineages as detailed in Methods. As expected, the TT-binding repertoire exhibited significantly higher mutation counts compared to the naive repertoire (S14 Fig). Additionally, the TT repertoire showed extensive clonal expansion, with 92% of sequences belonging to multi-cell lineages, compared to only 10% of sequences in naive mice, reflecting the different antigen exposure between the two repertoires. We found that antibodies from the TT repertoire exhibited larger paratope sizes compared to those from the naive repertoire (Fig 4C), suggesting that antigen-binding antibodies contain larger paratopes. Additionally, we noted a significant increase in paratope size in the mutated sequences relative to their germline ancestors in both the naive and TT repertoires (Fig 4D). This increase was particularly pronounced in the TT repertoire, where we observed that nearly all sequences had a larger paratope than their germline counterparts, suggesting that the process of SHM that leads to affinity maturation occurs through the creation of larger paratopes that enhance

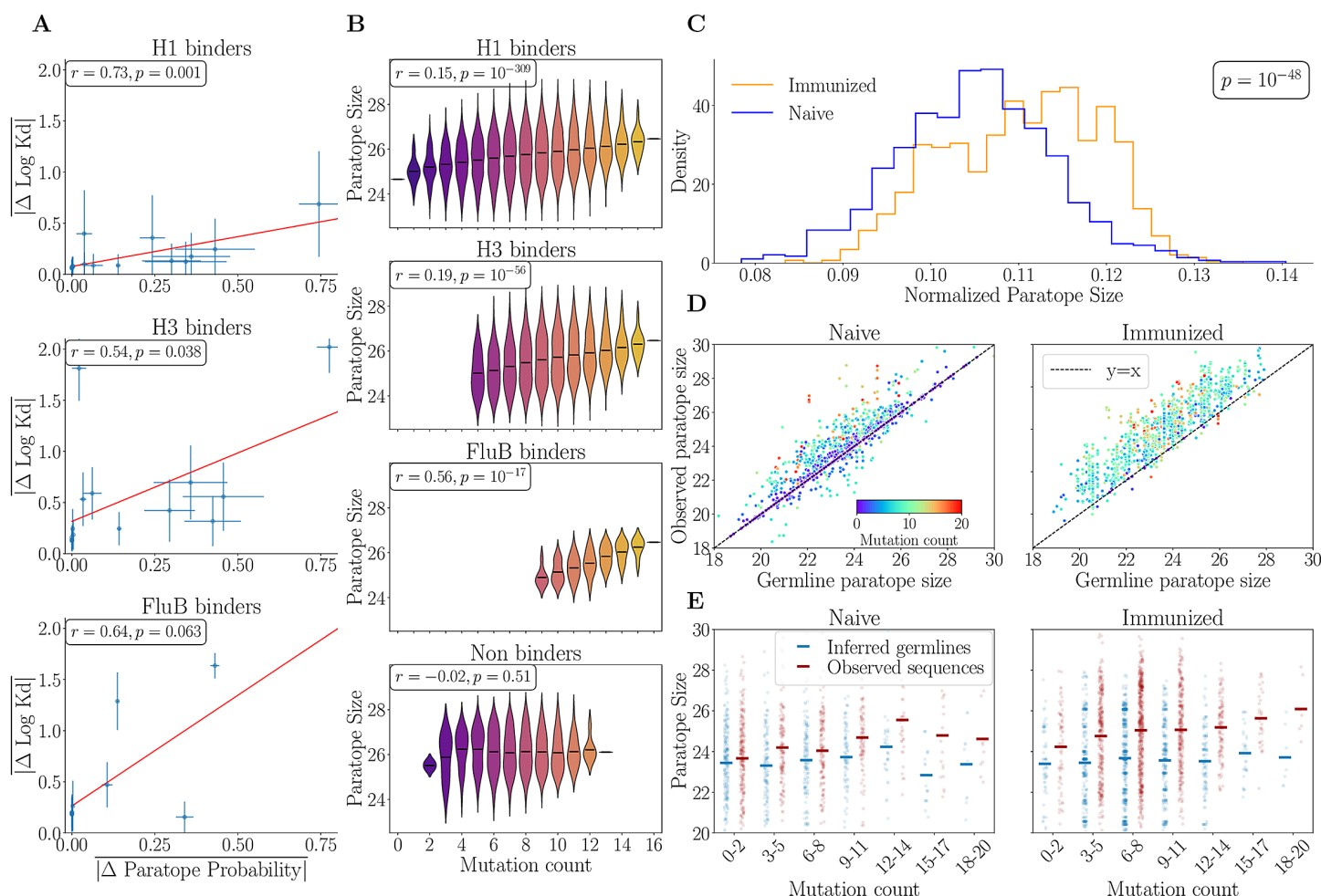

**Fig 4**. **(A) Correlation between the average change in affinity and the average change in the probability for an amino acid to belong to a paratope across the 16 mutated positions of bnAb 9114.** Averages are computed across all antibody variants with measurable affinity in [29] for each of the H1, H3, and FluB antigens. (B) Paratope size as a function of amino acid mutation count for three groups of binders and non-binders, based on experimental affinity measurements from [29]. Non-binders are defined as sequences with no measurable affinity to any of the three strains. For (A-B) linear correlation was quantified using Pearson's correlation coefficient (*r*), and the p-value computed with a two-sided hypothesis test (see pearsonr documentation). (C) Normalized paratope size density for a repertoire of IgG antibodies from mice immunized with tetanus toxoid [30] with antibodies sorted for binding to the antigen, compared to naive antibodies from the same mouse species [31]. The p-value was computed using a two-sided Mann–Whitney U test. (D) Comparison of paratope size between antibody sequences and their inferred germline sequences in the antibody repertoires of naive mice (left) and immunized mice (right). (E) Paratope size of observed antibody sequences and their germline sequences across different amino acid mutation count bins for naive mice (left) and immunized mice (right). The mutation count represents the number of amino acid differences between each antibody sequence and its germline, which is why germline sequences are also assigned mutation counts.

antigen binding. To ensure accurate mutation analysis, we preserved the observed CDR3 in our germline controls, focusing on V gene positions where germline inference is reliable. While this approach excludes speculative CDR3 mutations, it may create sequences that never existed in vivo. Additional analyses restricting to V genes or using the inferred germline D gene, and considering the most common V genes and CDR3 lengths, confirmed that the observed paratope increases were not biased by junctional variability, heavy V gene usage or CDR3 length (S15 Fig). Finally, we observed that in both repertoires, the paratope size increased with the number of mutations in the hypermutated sequences (Fig 4E). Importantly, this effect was not observed in the germline sequences themselves, indicating that the increase

in paratope size is a consequence of SHM rather than inherent differences in the original germline paratopes (Fig 4E). Notably, the effect of somatic mutations on paratope size was more pronounced in the immunized (TT) repertoire, suggesting that the mutations observed in antigen-binding antibodies were preferentially selected to enlarge the paratope and enhance antigen recognition. However, the correlations between paratope size and mutation count in the two repertoires ($r = 0.12$, $p = 1 \times 10^{-5}$ for the TT repertoire; $r = 0.24$, $p = 1 \times 10^{-16}$ for the naive repertoire) are not directly comparable due to differences in their mutation count distributions. To broaden our findings beyond the mouse immune system, and to showcase the ability of Paraplume to be used for extremely large bulk repertoires, we extended our analysis to a large healthy human antibody repertoire from Briney et al. [32]. Because our model maintains strong performance on heavy chains even with single-chain inputs (S2 Table), we applied the same methodology (cf. Methods) to this dataset, focusing specifically on IgG heavy chain sequences. After downloading the quality-processed reads from donor 316188, we retained approximately 4 million IgG sequences for analysis. Similarly to the mouse data, we found that the observed (affinity-matured) human heavy-chain sequences had larger paratope sizes compared to their germline counterparts (Fig 5A) and the paratope size correlated positively with the number of somatic mutations (Fig 5B, $r = 0.09$, $p < 10^{-16}$). However, the paratope size plateaued for sequences with more than 10 mutations (Fig 5B) suggesting the possibility that additional mutations may be neutral, increase affinity within the paratope without affecting its size, or work in a paratope-independent manner (e.g. by enhancing antibody stability). A similar early plateauing effect has also been reported in germinal center trajectory analyses, where most affinity gains occur within the first few mutations followed by a plateau and eventual decline, a phenomenon explained in part by survivorship biases [33].

Building on our comparison between immunized and naive repertoires, which suggested that antigen-binding antibodies tend to have larger paratopes, we sought to explore the relationship between selection and paratope size within a human repertoire lacking antigen-specific sorting. To distinguish more strongly selected antibodies from less selected ones, we used clonal lineage size as a proxy for positive selection, assuming that larger lineages reflect more successful affinity maturation and proliferation. For each lineage, we computed the average increase in paratope size of its sequences relative to their respective germlines. We observed that this average increase in paratope size positively correlates with lineage size (Fig 5C). Because larger lineages also tend to harbor more mutations, we controlled for mutation count and confirmed that the relationship between lineage size and paratope size increase remained robust (Fig 5D), indicating that paratope enlargement is associated with selection rather than mutation load alone. Interestingly, sequences bearing only one or two mutations show a paratope size decrease in small lineages (Figs 5D and S16), suggesting that such mutations may have had deleterious effects on the paratope that limited clonal expansion. Together, these findings suggest that paratope enlargement is more pronounced in lineages under stronger selection, likely reflecting additional rounds of affinity maturation and clonal expansion.

## Paratope-weighted sequence embeddings

We investigated whether incorporating paratope information could enable classification of binders versus non-binders, epitope class labeling, and improved prediction of antibody binding affinity. Protein language model embeddings are widely used to generate fixed-dimensional sequence representations, which serve as input to neural networks that predict binding affinity. A common approach involves averaging the embeddings of all amino acids in a sequence, thereby treating all residues equally, regardless of whether they belong to the framework region, complementarity-determining regions (CDRs), or the paratope. Building on the work of Ghanbarpour et al [34], we propose a sequence representation that weights amino acid embeddings based on their probabilities of belonging to the paratope, calculated by Paraplume (cf. Methods). In our analysis, we generate both unweighted and paratope-weighted representations for the six protein language models described in Methods and use them as inputs to a logistic regression model across two classification tasks—binder versus non-binder classification and epitope-class prediction (epitope binning). Details of the datasets, model, and evaluation procedure are provided in Methods. For clarity, we refer to these as unweighted embeddings and

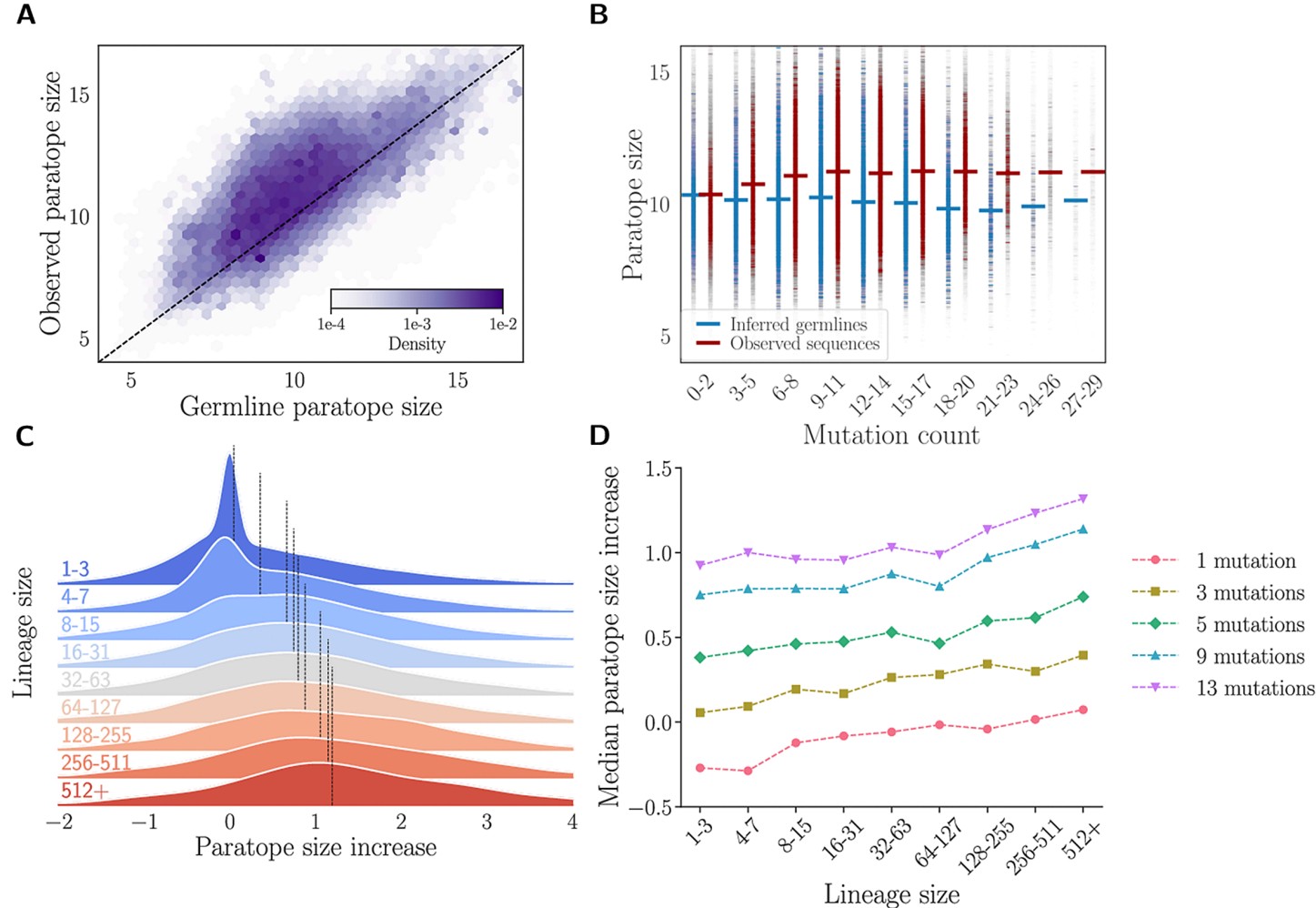

**Fig 5. Effect of hypermutations on paratope size in human repertoires.** Analysis for donor 326651 from [32]. (A) 2D histogram showing the relationship between the paratope size of observed antibody sequences and their inferred germline counterparts. (B) Paratope sizes of observed sequences and germline sequences grouped by amino acid mutation count bins. (C) Density of the average increase in paratope size within lineages, shown across different lineage size bins. Each density curve is fitted using all lineages in the corresponding size range. The black line indicates the median average increase in paratope size for each bin. (D) Median average increase when averaging over sequences with a fixed number of mutations within the lineage.

paratope-weighted embeddings, respectively. Across both tasks, paratope-weighted embeddings consistently outperformed unweighted embeddings (Fig 6), with the most significant gains observed when using ESM, a PLM not fine-tuned on antibodies. These improvements were statistically significant, as confirmed by a Wilcoxon paired sample test ($p = 0.007$ for binder classification; $p = 0.004$ for epitope binning). Next, we investigated whether incorporating paratope information improves binding affinity prediction using a linear regression model across three datasets with experimentally measured $K_D$ values (cf. Methods). Paratope-weighted embeddings produced modest and inconsistent changes in $R^2$ across models and datasets, with substantially worse performance for certain combinations of protein language models and datasets, and paired tests indicate that these differences are not consistently significant (S4 Table). As a negative control, we applied the same approach to predict antibody expression levels, a property unrelated to antigen binding; as expected, paratope-weighted embeddings offered no improvement over unweighted embeddings. These results highlight

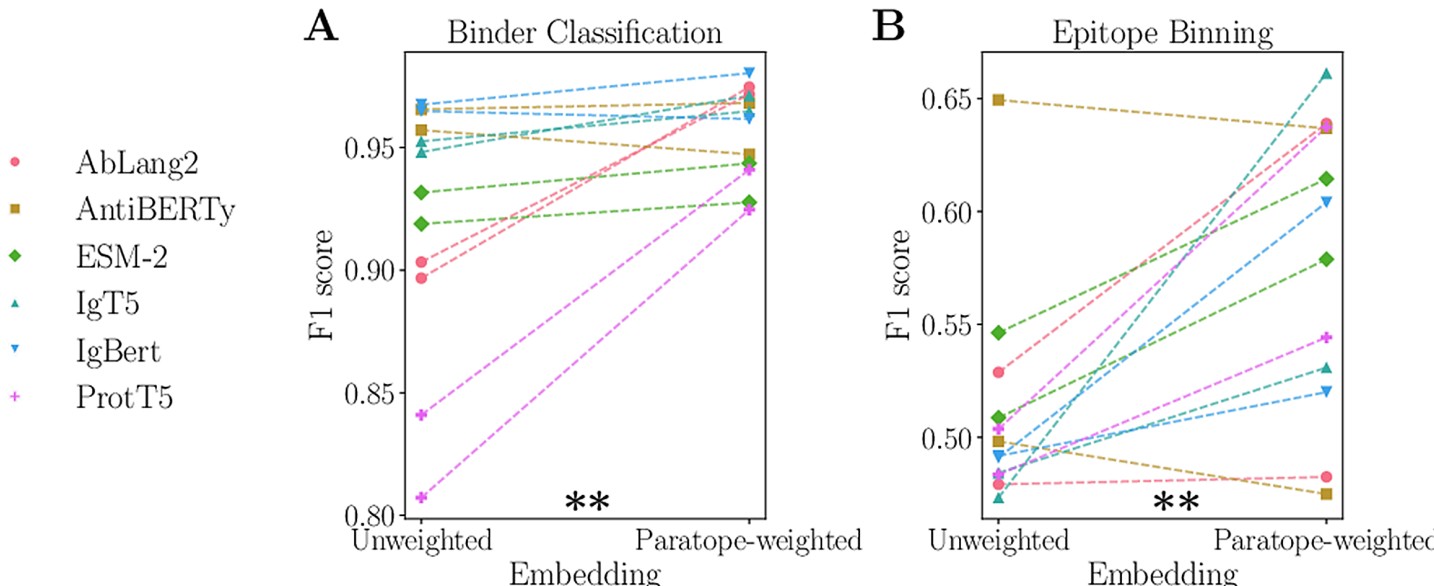

**Fig 6. (A) Comparison of the paratope-weighted and unweighted embeddings across the six large language models (LLMs) used in Paraplume.** Performance is evaluated using the F1 score from a regression model trained to classify binders versus non-binders, based on sequences from [29]. Two-fold cross-validation was performed on two distinct sets, resulting in the 12 data points. (B) The same analysis as in (A), but with a regression model trained to classify antibodies into epitope classes using sequences from [35]. A Wilcoxon paired sample test demonstrated that paratope-weighted embeddings yielded statistically significant improvements for both tasks, with p-values of 0.007 for binder classification and 0.004 for epitope binning.

that the benefits of paratope-weighted embeddings are context-dependent: they can be irrelevant for tasks unrelated to antigen binding, such as predicting antibody expression, and their impact on binding affinity prediction depends on the PLM and datasets used. Nevertheless, as shown in the classification tasks, paratope weighting can provide meaningful performance improvements when the task is closely related to antigen recognition, underscoring both the utility and the limitations of this approach.

## Discussion

Mapping the specific location of the paratope is important for both biotechnology and medicine. In therapeutic antibody design, accurate identification of antigen-binding residues enables engineering of binding properties through targeted mutations. Similarly, engineering therapeutic antibody developability often requires preserving paratope positions to avoid compromising binding function. Paratope residues are the most critical components of the antibody-antigen binding interface. Knowledge of how naive immunoglobulins evolve through the process of affinity maturation into effective antigen-specific antibodies, largely through expansion and change in paratope identity, is still poorly understood. Beyond individual antibodies, a rapid in silico paratope prediction method holds great promise for the large-scale analysis of affinity maturation in antigen-specific antibody repertoires. While sequencing technologies now allow high-throughput profiling of antibody repertoires, large-scale structural analysis remains challenging, as modeling thousands of antibodies is computationally demanding and often provides limited insight into the specific residues involved in antigen recognition.

Paratope prediction offers a scalable intermediate solution, bridging the gap between sequence-level data and functional interpretation. However, existing methods face several limitations that hinder their use in large-scale studies. Many rely on paired-chain inputs, restrict predictions to CDRs, or require structure prediction models, which limits throughput.

Paraplume's sequence-based and antigen-agnostic design offers a simpler and more scalable approach to studying mutational effects, eliminating the need for detailed structural modeling. We demonstrated that protein large language models can be used to develop a simple yet effective sequence-based paratope predictor. Paraplume avoids structural input dependencies, handles both paired or unpaired-chain data, and generalizes predictions across the full variable region. Despite its simplicity, Paraplume achieves performance on par with or exceeding that of current state-of-the-art sequence-based models on three different benchmark datasets. One limitation is that its MLP architecture is inherently opaque; to improve interpretability, we applied SHAP to quantify the contributions of individual PLMs across sequence regions. We note that more interpretable architectures, such as attention-based models, could be explored in future work to provide residue-level explanations.

Here, we highlight three conceptual advances of our work that go beyond incremental benchmarking or methodological improvements: the asymmetry-based analysis establishing a theoretical upper bound for sequence-based paratope prediction on the benchmark datasets used in this study, repertoire-scale analyses of antibody paratopes, and the use of paratope-weighted embeddings for downstream predictive tasks.

To better understand the biological limits of sequence-based paratope prediction, we leveraged the symmetry of antibody arms to estimate the intrinsic variability in paratope usage. Our analysis shows that this variability reflects genuine biological differences rather than technical artifacts, and that the variability of an antibody's paratope is strongly correlated with that of its cognate antigen epitope. We were able to use this variability to define a realistic upper bound for prediction accuracy, offering a useful reference point for evaluating current and future predictive models.

We further applied our model to investigate somatic hypermutation during antigen-driven immune responses and its influence on paratope identity. During affinity maturation within germinal centers, B cells undergo somatic hypermutation in the variable regions of both heavy and light chains of the B cell receptor. These mutations enhance the receptor's affinity and specificity for the target antigen. B cells that successfully navigate iterative cycles of mutation, selection, and clonal expansion ultimately differentiate into plasma cells or memory B cells, expressing antibodies with improved binding characteristics.

Our analysis reveals that affinity maturation in response to antigen exposure, which we measure through comparison of antigen-specific antibody sequences with their inferred germline state, is associated with an increase in predicted paratope size. This trend is particularly pronounced in clonally expanded antibody populations, indicating that enhanced antigen binding is a driving force behind this expansion. An expanded paratope allows for a greater number of chemically compatible interactions with the cognate epitope, thereby increasing binding affinity. Moreover, the requirement for additional interacting residues inherently demands a broader epitope interface, which in turn contributes to enhanced antibody specificity. This finding opens up the possibility of using changes in predicted paratope size as a proxy for increased antigen-specificity and affinity. This could be particularly useful for in silico methods of affinity maturation to predict those changes that will increase affinity and those that won't.

Another practical application of this work is the use of paratope-weighted embeddings whereby incorporating paratope information in PLM embeddings can enhance fine-tuning of models trained for antibody functional prediction. These paratope-weighted embeddings consistently outperform standard averaged embeddings in classification tasks associated with antibody function such as binder classification or epitope binning. However, they do not consistently outperform averaged embeddings for predicting binding affinity. One limitation is that residues distal to the paratope contribute less to the sequence embedding, even though they can influence binding through allosteric effects. In addition, the datasets available for affinity prediction are limited in diversity, which reduces the ability to assess model generalization. This work challenges the assumption that structural modeling is essential for studying antibody-antigen interactions and instead positions PLM-driven sequence-based paratope prediction as a powerful, scalable tool for repertoire-level analyses. In doing so, it opens new avenues for exploring the functional consequences of antibody diversification and evolution.

Looking ahead, several components of our model stand to benefit from ongoing advancements. The continuous growth of structural antibody databases like SAbDab will enable training on larger and more diverse datasets. Simultaneously,

improvements in protein language models, driven by increasing availability of sequence data and advances in representation learning, will enhance the quality of input embeddings. Future work should aim to integrate these developments, with the goal of further improving paratope prediction accuracy and extending its applications in large-scale repertoire analysis, therapeutic antibody affinity and developability engineering and generative antibody creation.

## Methods

### Protein large language models embeddings

ESM-2 and ProtTrans are protein large language models (PLMs), whereas Antiberty is an antibody-specific model pretrained on 558M natural antibody sequences. For ESM-2, we chose the pretrained model with 33 layers and 650 million parameters, denoted as `t_33_650M`. IgT5 and IgBert are PLMs fine tuned on antibody sequences, and derive their names from the well-known NLP models T5 [36] and BERT [37]. One key difference between the two is that BERT predicts a single masked token at a time, whereas T5 does not have a predefined number of masked tokens to predict. To address the bias introduced by the predominance of germline-encoded residues in antibody sequences, Olsen et al. [18] developed Ablang2, a model optimized for the prediction of mutated residues. We assess the contribution of each of the six PLMs by comparing model performance under three settings: using all embeddings, using individual embeddings, and using all embeddings except one (S1 Table). Across the three benchmark datasets and four evaluation metrics, removing any single embedding led to a performance drop in at least one dataset, highlighting the complementarity of the six models.

### Loss function

To train our model, we use the Binary Cross Entropy (BCE) loss function. It quantifies the difference between the model's predicted probability outputs and the true binary labels and is defined as:

$$\text{BCE} = -\frac{1}{N}\sum_{i=1}^{N}\left(y_i\log(p_i) + (1 - y_i)\log(1 - p_i)\right),$$

where $N$ is the number of samples, $y_i$ is the true label for the $i$-th sample (either 0 or 1), and $p_i$ is the predicted probability for the $i$-th sample. By minimizing the BCE loss during training, the model learns to output paratope probabilities that closely match the true labels for each amino acid, thereby improving its classification performance.

### Training protocol

Training incorporates several standard regularization techniques, including dropout with a rate of 0.4 applied to all hidden layers, random masking of 40% of the input embeddings, and early stopping with a patience of 10 epochs and a maximum of 300 epochs. Hyperparameters were selected through a grid search on the Paragraph dataset and subsequently fixed across all three benchmark datasets. The final configuration includes a learning rate of $1 \times 10^{-5}$, a batch size of 16 sequences, an ADAM optimizer with L2 regularization weight of $1 \times 10^{-5}$, and 3 hidden layers of widths 2000, 1000 and 500 for the MLP. A summary of the explored hyperparameter ranges and the chosen values is provided in S5 Table. All experiments were performed using Paraplume version 1.0.0 on a workstation equipped with two NVIDIA RTX 5000 Ada GPUs (32 GB VRAM). Models were trained and evaluated with seeds 1 through 16, and the reported results correspond to averages across all seeds. Scripts and documentation to reproduce the results are available on the github page (cf. Methods at data and code availability).

## Single chain Vs Paired chain

Some PLMs are designed to process individual chains (ESM-2, ProtTrans, Antiberty), while others (IgBert, IgT5, AbLang2) are fine-tuned on paired antibody chains and can handle either paired or single chains. Paraplume supports both single-chain and paired-chain modes, depending on how the input embeddings are generated. When both heavy and light chains are available, Paraplume generates embeddings by concatenating the two chains for models that operate on single sequences (e.g., ESM-2, ProtTrans, Antiberty), and passing each chain separately to models fine-tuned on paired chains (e.g., IgBert, IgT5, AbLang2). In single-chain mode, only one chain (heavy or light) is used. Embeddings are computed using each PLM's single-chain version, including those normally fine-tuned on paired data. Thus, the distinction between the single and paired versions of Paraplume lies solely in how the embeddings are generated, not in the model architecture itself. In S2 Table we compare the two settings by evaluating Paraplume's performance separately on heavy and light chains across the benchmark datasets.

## PLM ablation and complementarity benchmark

To quantify how individual protein language models (PLMs) contribute to paratope prediction and to assess whether their embeddings provide complementary information, we performed an ablation study across the six PLMs used in Paraplume.

The benchmark consists of six stages, corresponding to concatenation sets of size $i \in \{1, \dots, 6\}$. Rather than evaluating all $\binom{6}{i}$ combinations, each stage extends a selected configuration from the previous stage. In Stage 1, we use ESM because it performs best across all metrics. In Stage 2, we pair this Stage 1 PLM with each of the five remaining PLMs, yielding five two-PLM models. Following stages adopt the same approach, progressively adding PLMs until Stage 6, which evaluates all six models. This sequential approach probes complementarity while keeping the number of models trained manageable.

In all experiments, residue-level embeddings from selected PLMs were concatenated and passed to a fixed MLP classifier. However this does not ensure that performance differences arise solely from the choice and combination of embeddings. To control for increases in input dimensionality, each stage also includes two capacity-matched baselines. Specifically, we construct models where the ESM embedding is concatenated with itself $i$ times, and likewise where the ProST-T5 embedding is repeated $i$ times. These baselines preserve the expanded input size but remove embedding diversity, allowing us to attribute performance gains directly to PLM complementarity rather than increased model capacity.

All models are trained using Paraplume 1.1.0 with a fixed learning rate of $5 \times 10^{-5}$ to accelerate training. Results are averaged across eight random seeds for robustness. For every configuration, we report PR AUC, F1 score, Matthews correlation coefficient for classification (MCC), parameter count, peak GPU memory usage, and inference throughput. All experiments were run on a workstation equipped with two NVIDIA RTX 5000 Ada GPUs (32 GB VRAM), with each run assigned to a single GPU.

## Shapley-value–based interpretability

To interpret the contribution of each protein language model (PLM) within Paraplume, we compute Shapley values over the input features of the model. Shapley values quantify how much each feature contributes to the difference between the prediction with all features present and the prediction under any subset of features [21]. In this setting, the features are the dimensions (scalar components) from all PLMs for a given residue, and the value function $f$ corresponds to the MLP's paratope probability prediction.

Formally, for a feature set $F$ and feature $i \in F$, the Shapley value is

$$\phi_i = \sum_{S \subseteq F \setminus \{i\}} \frac{|S|!\,(|F| - |S| - 1)!}{|F|!} \left[ f(S \cup \{i\}) - f(S) \right],$$

where $f(S)$ denotes the model prediction when only the features in subset $S$ are provided. This expression represents the expected marginal contribution of feature $i$ over all possible coalitions.

We propose two different interpretability measures at the PLM level, available as part of the software through visualization options. First, we compute raw Shapley importance scores for each PLM by summing the Shapley values of all its embedding dimensions. Second, to control for differences in embedding dimensionality, we propose a mean Shapley importance by averaging a PLM's Shapley values over its dimensions. We then normalize each set of PLM importance scores to sum to 1,

$$\tilde{I}_k = \frac{I_k}{\sum_j I_j},$$

where $I_k$ denotes the summed or mean Shapley score for PLM $k$.

## A dataset to analyze paratope asymmetry

To analyze paratope asymmetry we keep PDB files from the SabDab database [17] meeting the following criteria, as shown Fig 3A: (1) the antigen must be a peptide or protein; (2) the antibody must consist of a heavy and light chain, thereby excluding nanobodies; (3) the antibody must have exactly 2 side chains; (4) the two antibody side chains must be identical (i.e., the paired heavy-chain sequences match and the paired light-chain sequences match), as well as the two antigen sequences they bind, therefore removing the risk to analyze antibodies engineered extensively to be bi-specific or containing missing residues. Following this set of filters, a total of 470 antibodies were retained for analysis. The metadata includes one row per antibody-antigen-interaction, describing one heavy and light chain of the antibody bound to an antigen chain.

## Paratope and epitope asymmetry

The paratope asymmetry between the paratopes of two identical antibody arms is defined as the count of all amino acids present in one of the two paratopes, but not in both. Given two paratopes $P_1 = \{a_{\text{pos}_1}, \dots, a_{\text{pos}_n}\}$ and $P_2 = \{b_{\text{pos}_1}, \dots, b_{\text{pos}_m}\}$, where $a_{\text{pos}_i}$ (respectively $b_{\text{pos}_j}$) represents the amino acid at position $\text{pos}_i$ ($\text{pos}_j$) in the sequence, this can formally be written as a symmetric difference:

$$\text{Card}\left((P_1 \cup P_2) \setminus (P_1 \cap P_2)\right)$$

For example, for two paratopes $\{L_{63}, Q_{64}, G_{66}\}$, $\{L_{63}, Q_{64}, A_{67}\}$ the asymmetry is $\text{Card}(\{G_{66}, A_{67}\}) = 2$ The normalized paratope asymmetry is then

$$\frac{\text{Card}\left((P_1 \cup P_2) \setminus (P_1 \cap P_2)\right)}{\text{Card}(P_1 \cup P_2)}.$$

which corresponds to the Jaccard distance $d_J$

$$= 1 - \frac{\text{Card}(P_1 \cap P_2)}{\text{Card}(P_1 \cup P_2)} = 1 - J(P_1, P_2) = d_J(P_1, P_2)$$

A high asymmetry is close to 1, whereas a low asymmetry is close to 0.

Epitope asymmetry and normalized epitope asymmetry are defined analogously using the epitopes of the two identical antigens bound to each of the antibody's arms.

## Antibody repertoire analysis

Germline versions of each antibody were generated by identifying the closest V and J germline genes using IgBlast [38]. The V and J regions of each antibody were then replaced with the inferred germline sequences, while retaining

the original CDR3 sequences in both heavy and light chains due to the difficulty of accurately inferring germline CDR3 regions. This approach allowed for paratope prediction across the entire variable region. Somatic hypermutations (SHMs) were defined as the number of amino acid differences between the original antibody sequences and their corresponding inferred germline counterparts. Lineages were inferred using HILARy [39], which offers high precision and minimizes erroneous clustering of antibodies coming from distinct lineages. We predicted the paratopes of all antibodies as well as their germline ancestors with Paraplume trained on the complete expanded dataset from [9]. We then checked that none of the antibody datasets analyzed shared high sequence similarity with the data used to train Paraplume, thereby avoiding train-test leakage (S5B Fig).

## Unweighted Vs paratope-weighted embedding

Let a sequence of amino acids be represented as a set of embeddings:

$$\mathbf{E} = \{\mathbf{e}_1, \mathbf{e}_2, ..., \mathbf{e}_N\}, \quad \mathbf{e}_i \in \mathbb{R}^d,$$

where $\mathbf{e}_i$ is a $d$-dimensional embedding of the $i$-th amino acid in a sequence of length $N$.

The standard approach for sequence representation is to compute the unweighted mean of all amino acid embeddings:

$$\mathbf{e}_{\text{avg}} = \frac{1}{N} \sum_{i=1}^{N} \mathbf{e}_i.$$

To integrate paratope information, we compute a weighted average of the amino acid embeddings, where the weights are derived from the normalized paratope probabilities $p_i$, representing the likelihood that the $i$-th amino acid is part of the paratope, as predicted by Paraplume:

$$\mathbf{e}_{\text{para}} = \sum_{i=1}^{N} w_i \mathbf{e}_i, \quad \text{where} \quad w_i = \frac{p_i}{\sum_{j=1}^{N} p_j}.$$

## Antibody classification

For the binder classification task, we used data from Phillips et al. [29], selecting 111 high-affinity antibodies targeting the FluB strain and pairing each with a low-affinity mutant differing by one residue. We fit a logistic regression model using sequence embeddings (paratope-weighted or unweighted) to predict binder status, using cross-entropy loss and evaluating performance via F1-score. For epitope binning, we curated a dataset from CoV3D [35] comprising 329 antibodies targeting the SARS-CoV-2 RBD, grouped into four epitope classes. We applied a one-vs-rest logistic regression framework, training one binary classifier per epitope group. The average F1-score across classes was used to assess overall performance. For both tasks, datasets were split into two equal, non-overlapping subsets. Two-fold cross-validation was performed within each subset, across six PLMs, resulting in 12 evaluations per task and embedding type. Results were averaged over five random seeds to ensure robustness. Statistical comparisons between embedding strategies were conducted using the Wilcoxon paired sample test.

## Binding affinity prediction

We followed the methodology of Kenlay et al. [20], using three datasets from [40], [41], and [42], containing 422, 2048, and 4275 antibody sequences, respectively, each paired with $K_D$ measurements against a target antigen. For each dataset, we applied regularized linear regression to predict $\log(K_D)$ from either unweighted or paratope-weighted embeddings, using 10-fold cross-validation. Model performance was evaluated using the coefficient of determination $R^2$ on the

test sets. To validate task-specific relevance of paratope information, we applied the same method to predict antibody expression levels using data from [42].

## Supporting information

**S1 Fig. Comparison of inference time and $CO_2$ emissions for Paragraph, Paraplume, and Paraplume-S.** Inference time was compared across different numbers of sequences on an NVIDIA RTX 5000 Ada Generation GPU (A) and 96-core Intel(R) Xeon(R) Gold 6442Y CPUs (B). For Paragraph, 3D structures were generated using AbodyBuilder3, the fastest available structure prediction tool to our knowledge, to ensure a fair comparison. We also compared $CO_2$ emissions using the package codecarbon [43], on GPU (C) and CPU (D).
(TIFF)

**S2 Fig. Ablation study showing performance and computational cost trade-offs of individual PLMs.** Incremental changes in performance (top row) and computational cost (bottom row) as additional PLMs are sequentially incorporated into the model. All experiments were run on a workstation equipped with two NVIDIA RTX 5000 Ada GPUs (32 GB VRAM), with each run assigned to a single GPU. The PR AUC, F1 score, and Matthews correlation coefficient (MCC) are the evaluation metrics also used in our benchmark with other methods. F1 and MCC are computed using a threshold of 0.5. Results are averaged over 8 seeds and obtained using Paraplume 1.1.0.
(TIFF)

**S3 Fig. Sequence visualization: Paraplume predictions and raw PLM importance scores (Shapley values).** The shown sequence corresponds to the heavy chain of PDB complex 1BVK from the Paragraph test set, with Paraplume trained on the Paragraph training set.
(TIFF)

**S4 Fig. Visualization of raw PLM importance scores at the dataset level.** Scores are computed using Shapley values (Shapley-Value–Based Interpretability) and averaged across amino acids within each sequence to obtain a single score per sequence. Complexes shown are from the Paragraph test set with Paraplume trained on the Paragraph training set.
(TIFF)

**S5 Fig. Sequence similarity distributions comparing benchmark datasets, antibody repertoires, and Paraplume training data.** (A) Histogram of sequence similarity between the three benchmark training and test sets. (B) Histogram of sequence similarity between the four analyzed antibody repertoires and the Paraplume training dataset.
(TIFF)

**S6 Fig. Paratope ground truth versus Paraplume predictions on unseen antibody, full variable and framework regions.** (A) Comparison of ground truth paratope labels (left) and Paraplume model predictions (right) for the full variable region of the 6B0S antibody-antigen complex, which was not included in the training set. For visualization, antibodies were depicted as spheres and the antigen in a cartoon representation (green) in PyMOL [44]. In the ground truth structure, residues forming the paratope are highlighted in red. The colorbar shows the probability of a given amino acid being a paratope residue. For clarity, only the $C_\alpha$ carbon of each residue is depicted. (B) Same structure as in (A) but restricted to amino acids belonging to the framework region.
(TIFF)

**S7 Fig. Statistics of the dataset curated to study paratope asymmetry (470 antibody-antigen complexes).** Histograms of the (A) paratope asymmetry and (B) paratope size. (C) Heatmap of paratope asymmetry against paratope size, colored by number of sequences. $r$ is the Pearson correlation coefficient. (D) Histogram of the paratope asymmetry normalized by the paratope size. (E-H) Same as (A-D) but for the epitope.
(TIFF)

**S8 Fig. Normalized paratope and epitope asymmetry against PDB characteristics (methods, antigen type, resolution).** Violin plots of the normalized paratope asymmetry separated by crystallography method (A) and antigen type (B). Normalized paratope asymmetry against PDB resolution (C). (D-F) Same but for the normalized epitope asymmetry.
(TIFF)

**S9 Fig. Sensitivity checks: normalized paratope asymmetry vs. arm/antigen differences.** (A-B) Normalized paratope asymmetry as a function of antibody arm difference (A) and antigen sequence difference (B), computed using Levenshtein distance. (C-D) Normalized paratope asymmetry as a function of normalized epitope asymmetry for antibodies with arm differences between 1-10 (C) and 10-20 (D). (E-H) Same analysis for fixed arm sequence differences ranging from 1 (E) to 4 (H). Pearson correlation coefficient ($r$) is reported.
(TIFF)

**S10 Fig. Arm-to-arm upper bound distribution of F1 and MCC scores.** Distribution of arm-to-arm F1 (A) and MCC (B) scores on the sequences from Paragraph test set.
(TIFF)

**S11 Fig. Ambiguous residue analysis: Paraplume predictions for consistent and ambiguous residues.** Comparison of Paraplume's predictions for consistent residues (same paratope label in both arms) and ambiguous residues (different paratope labels in both arms) in the PECAN dataset (A) and Paragraph dataset (B).
(TIFF)

**S12 Fig. Paratope size proxy analysis and comparison with AlphaFold Multimer v3.** (A) Correlation of paratope size computed using Paraplume with the buried Solvent Accessible Surface Area (bSASA) of ground truth crystal structures of antibody-antigen pairs for all protein antigens of the Paragraph test set, with Paraplume trained on the Paragraph training set. (B) Correlation of the bSASA of the structures predicted using AlphaFold-Multimer and the bSASA of the ground truth crystal structures. Structures were predicted using ColabFold default settings and AlphaFold-Multimer v3. Linear correlation was quantified using Pearson's correlation coefficient $r$, and the p-value computed with a two-sided hypothesis test (see pearsonr documentation).
(TIFF)

**S13 Fig. Correlation between thresholded paratope counts and buried Solvent Accessible Surface Area (bSASA).** Correlation between thresholded counts of paratope amino acids predicted by Paraplume (probability thresholds: 0.25, 0.5, and 0.75 for A, B, and C panels, respectively) and the buried solvent-accessible surface area (bSASA) from crystal structures of antibody-antigen pairs for all protein antigens of the Paragraph test set, with Paraplume trained on the Paragraph training set. Linear correlation was quantified using Pearson's correlation coefficient $r$, and the p-value computed with a two-sided hypothesis test (see pearsonr documentation).
(TIFF)

**S14 Fig. Mutation count distributions for the naive and immunized mouse repertoires.** Amino acid mutation count distribution for (A) the immunized mouse antibody repertoire of [30] and (B) the naive mouse antibody repertoire of [31].
(TIFF)

**S15 Fig. Paratope size control for the immunized mouse antibody repertoire.** Controls for changes in paratope size between observed IgG sequences and their inferred germlines in mice immunized with tetanus toxoid [30]. Paratope size change is defined as Δ paratope size (observed sequence minus germline). (A) Distribution of Δ paratope size under three CDR3 handling strategies for germline reconstruction: (1) retaining only the V gene (CDR3 and J removed); (2) inclusion of the most likely germline D gene; and (3) use of the observed CDR3 sequence. (B–C) Δ paratope size stratified by the five most frequent V genes (B) and by the five most frequent CDR3 lengths (C).
(TIFF)

**S16 Fig. Average paratope size increase across different lineage sizes.** The average in computed over sequences with a fixed number of mutations within the lineage (from $N = 1$ top left to $N = 16$ bottom right). Each point is a lineage, and the mean average increase is the thick line.
(TIFF)

**S1 Table. Ablation study: impact of PLM embedding configurations on performance across three datasets.** The default Paraplume configuration uses all six embeddings: AbLang2, antiBERTy, IgBert, IgT5, ESM-2, and ProtT5. We report results when each embedding is removed individually or used alone. Bold indicates the best score, and underlined values represent the second-best. Based on performance across all 12 evaluation points, we chose to retain all six embeddings. While choosing different settings for each dataset could yield higher scores, we prioritize robustness and use the same configuration across all three datasets. Paraplume-S is a lightweight variant of Paraplume that uses only ESM-2 embeddings. All results are averaged over 16 random seeds to account for variability.
(XLSX)

**S2 Table. Comparison of Paraplume using paired and single chain embeddings.** We evaluate Paraplume's performance on individual chains (heavy or light, as indicated in the Test chain column) by comparing two input settings: embeddings generated from both chains (Paraplume-Paired) versus embeddings generated from only the test chain (Paraplume-Single). Results show that using heavy chain embeddings leads to a slight but acceptable drop in performance compared to using paired chain embeddings, indicating that Paraplume remains robust even in the absence of paired chain information.
(XLSX)

**S3 Table. Comparison of Paraplume and Paragraph across different regions of the antibody sequence.** Paragraph-ABB refers to Paragraph using structures modeled with ABodyBuilder, while Paragraph-crystal refers to Paragraph trained on experimentally determined structures. Results for Paragraph-ABB are taken from the original study [9], whereas results for Paragraph-crystal were computed by retraining Paragraph on experimentally-determined structures.
(XLSX)

**S4 Table. Comparison of methods for generating sequence embeddings.** The paratope-weighted embedding is computed as a weighted average of amino acid embeddings, with weights determined by their predicted probabilities of belonging to a paratope, while the averaged embedding is a uniform mean across all amino acid embeddings. Performance is assessed using the $R^2$ score of a linear model predicting binding affinity or expression using the sequence embedding as input, across different protein language models (PLMs) and datasets. A paired Wilcoxon test across the six PLMs and three datasets (18 points) shows that paratope weighting does not provide a statistically significant improvement for affinity prediction ($P = 0.393$).
(XLSX)

**S5 Table. Summary of hyperparameters explored, their ranges, and optimal values on the Paragraph dataset.**
(XLSX)

## Author contributions

**Conceptualization:** Gabriel Athènes, Adam Woolfe, Thierry Mora, Aleksandra M. Walczak.

**Data curation:** Gabriel Athènes.

**Formal analysis:** Gabriel Athènes, Adam Woolfe, Thierry Mora, Aleksandra M. Walczak.

**Funding acquisition:** Adam Woolfe, Thierry Mora, Aleksandra M. Walczak.

**Investigation:** Gabriel Athènes, Adam Woolfe, Thierry Mora, Aleksandra M. Walczak.

**Methodology:** Gabriel Athènes, Adam Woolfe, Thierry Mora, Aleksandra M. Walczak.

**Project administration:** Adam Woolfe, Thierry Mora, Aleksandra M. Walczak.

**Resources:** Gabriel Athènes, Adam Woolfe, Thierry Mora, Aleksandra M. Walczak.

**Software:** Gabriel Athènes.

**Supervision:** Adam Woolfe, Thierry Mora, Aleksandra M. Walczak.

**Validation:** Gabriel Athènes.

**Visualization:** Gabriel Athènes.

**Writing – original draft:** Gabriel Athènes, Adam Woolfe, Thierry Mora, Aleksandra M. Walczak.

**Writing – review & editing:** Gabriel Athènes, Adam Woolfe, Thierry Mora, Aleksandra M. Walczak.

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
