## [Decision Letter · Decision Letter 0]

14 Nov 2025

PCOMPBIOL-D-25-02136

Paraplume: A fast and accurate paratope prediction method provides insights into repertoire-scale binding dynamics

PLOS Computational Biology

Dear Dr. Walczak,

Thank you for submitting your manuscript to PLOS Computational Biology. After careful consideration, we feel that it has merit but does not fully meet PLOS Computational Biology's publication criteria as it currently stands. Therefore, we invite you to submit a revised version of the manuscript that addresses the points raised during the review process.

We look forward to receiving your revised manuscript.

Kind regards,

Lun Hu

Academic Editor

PLOS Computational Biology

Dominik Wodarz

Section Editor

PLOS Computational Biology

**Additional Editor Comments:**

All reviewers see value in this work, but they raised several critical concerns, such as in-depth analysis of experimental results, extension of ablation studies, and interpretability. Authors should carefully revise their manuscript by thoroughly addressing all these issues.

**Journal Requirements:**

At this stage, the following Authors/Authors require contributions: Gabriel Athènes, Adam Woolfe, Thierry Mora, and Aleksandra M. Walczak. Please ensure that the full contributions of each author are acknowledged in the "Add/Edit/Remove Authors" section of our submission form.

4) Your manuscript is missing the following sections: Abstract.  Please ensure all required sections are present and in the correct order. Make sure section heading levels are clearly indicated in the manuscript text, and limit sub-sections to 3 heading levels. An outline of the required sections can be consulted in our submission guidelines here:

5) Please upload all main figures as separate Figure files in .tif or .eps format. For more information about how to convert and format your figure files please see our guidelines:

6) We notice that your supplementary Figures, and Tables are included in the manuscript file. Please remove them and upload them with the file type 'Supporting Information'. Please ensure that each Supporting Information file has a legend listed in the manuscript after the references list.

7) Please amend your detailed Financial Disclosure statement. This is published with the article. It must therefore be completed in full sentences and contain the exact wording you wish to be published.

State the initials, alongside each funding source, of each author to receive each grant. For example: "This work was supported by the National Institutes of Health (####### to AM; ###### to CJ) and the National Science Foundation (###### to AM).".

**Reviewers' comments:**

Reviewer's Responses to Questions

**Comments to the Authors:**

Reviewer #1: The review is uploaded as an attachment

Reviewer #2: ## Summary

The paper introduces Paraplume, a novel computational method for antibody paratope prediction. Unlike approaches that rely on 3D structures, Paraplume is purely sequence-based. Its architecture leverages embeddings from six different Protein Language Models (PLMs). These embeddings are concatenated and fed into a MLP to predict the paratope probability for each amino acid.

The method was validated on three benchmark datasets (PECAN, Paragraph, MIPE), demonstrating performance that is competitive with or exceeds existing structure-based methods.

## Strength

1. **Scalability:** The method's primary strength is its computational efficiency. By avoiding 3D structure modeling, Paraplume is a highly scalable tool practical for analyzing antibody repertoires at a scale of millions of sequences.

2. **Strong Empirical Performance:** Despite its architectural simplicity, Paraplume demonstrates excellent performance, notably outperforming methods that rely on *modeled* 3D structures. This provides strong support for the utility of rich sequence embeddings.

## Major Points

While the work has clear merits, several key issues regarding the methodology and experimental validation must be addressed.

1. **Lack of Cost-Benefit Analysis for PLM Embeddings:** The paper claims "complementarity" among the six PLMs but only provides a cost analysis for the final 6-PLM (Paraplume) vs. 1-PLM (Paraplume-S) models. The marginal performance gain of each individual PLM is not weighed against its specific computational cost (e.g., inference time, VRAM usage).

2. **Confounding Variable: Model Capacity vs. Informational Synergy:** The core methodological claim is confounded. Concatenating six embeddings does not just add "complementary information"; it massively increases the input dimensionality and, consequently, the total parameter count of the MLP (especially in the first hidden layer). The observed performance gain may simply be an artifact of this increased model capacity, rather than true informational synergy.

3. **Inadequate Ablation Studies:** Following Point 2, the ablation studies (Table S1) are insufficient. They compare a 6-PLM model to 1-PLM models *with the same MLP architecture*. A proper control is missing: a 1-PLM model (e.g., Paraplume-S) fed into a "wider" or "deeper" MLP with a parameter count equivalent to the full Paraplume model. Without this control, the central claim of "complementarity" is unsubstantiated.

4. **Omission of Key Model Comparisons:** The paper omits comparisons to a critical third class of models: structure-aware sequence models (e.g., ProstT5, TM-vec). These models explicitly integrate 3D knowledge during pre-training but require only sequence input at inference. To validate the claim that Paraplume's *implicit* structural information is superior, it must be benchmarked against these *explicit* structure-aware SOTA methods.

5. **Insufficient Case Study:** The case study in Figure 2 demonstrates Paraplume's success but not its *superiority*. A more compelling case would compare Paraplume against a structure-based method (e.g., Paragraph-ABB) on an antibody where the underlying 3D structure model (e.g., ABodyBuilder) is known to fail (e.g., a complex CDR loop), showing that Paraplume succeeds where the structure-based method fails.

6. **Lack of Model Interpretability:** The model is treated as a "black box." The claim of "complementarity" is an untested hypothesis. The authors should apply feature attribution methods (e.g., SHAP, Integrated Gradients) to the trained MLP. This would reveal which PLMs the model relies on for different predictions (e.g., CDR3 vs. framework regions) and provide mechanistic evidence for the complementarity claim.

7. **Omission of Interpretable Architectures:** The choice of an MLP, while simple, sacrifices model interpretability. Other architectures, such as those using attention mechanisms (e.g., VenusVaccine), provide clear, residue-level explanations for their predictions. The authors should justify why an opaque MLP was chosen over more transparent alternatives and demonstrate some aspect of their model's interpretability.

## Minor Points

1. **Incorrect Terminology in Figure:** In Figure 1B, the label "Pretrained LLMs" is used, maybe "Pretrained PLMs"?

2. **Figure Quality:** Figures 1 and 2 require improvement. Figure 1B is overly simplistic; a more informative schematic might briefly illustrate the different types of PLMs used. Figure 2 is a basic render; a more aesthetic visualization would be more impactful.

3. **Accessibility of Citations:** The submitted PDF lacks active hyperlinks for citations.

4. **Critical Link Error:** The Zenodo link provided on Page 4 (.../records/17277448) contradicts the link provided on Page 18 (.../records/17021232).

Reviewer #3: This manuscript presents Paraplume, a fast and accurate paratope prediction framework that enables repertoire-scale analyses of antibody binding, and it demonstrates solid innovation, good readability, and a rich, convincing set of experiments. The following points require clarification:

1.The Introduction and Discussion could more clearly articulate what is conceptually new relative to existing paratope predictors such as Paragraph, PECAN, MIPE and Parapred. In particular, please spell out which aspects are the true contributions (multi-PLM sequence-only paratope prediction, paratope-weighted embeddings, repertoire-scale analyses, asymmetry-based upper bound) and which are incremental improvements in benchmarking.

2.The manuscript would benefit from a more detailed description of how sequence redundancy and potential train–test leakage are controlled across the PECAN, Paragraph, MIPE, and expanded SAbDab datasets (e.g., sequence-identity thresholds, overlap between ABB3 / AlphaFold training data and the benchmark splits, and between Paraplume’s training set and the repertoires analyzed downstream). Given that you already note ABB3 training/test overlap for Paragraph on MIPE, a systematic treatment of these issues is important for a fair comparison.

3.While Section IV. A briefly justifies the use of six PLMs and mentions an ablation in Table S1, the main text does not fully convey how much each embedding source contributes versus the added computational cost. I suggest moving at least a subset of these ablation results into the main Results, and clarifying whether a learned linear combination or dimensionality reduction of concatenated embeddings was tested as an alternative to the very high-dimensional concatenation.

4.The analysis of paratope/epitope asymmetry in antibodies binding identical antigens is interesting, but the resulting ~0.95 F1 “upper bound” is presented somewhat strongly as a performance ceiling. Please clarify that this bound depends on the chosen distance cutoff, labeling protocol, and dataset composition, and consider showing the full distribution of arm-to-arm F1/PR AUC across complexes to emphasize variability rather than a single number.

5.The conclusion that somatic hypermutation leads to larger paratopes in mouse and human repertoires is compelling but may be influenced by confounders such as total variable-region length, CDR3 length, or V-gene usage. Since “paratope size” is defined as a sum over per-residue probabilities, it should be made clearer how you control for length and composition (e.g., via normalized paratope size, length-stratified analyses, or multivariable models including SHM count and CDR length).

6.The Methods state that hyperparameters were selected on the Paragraph dataset and then reused for all benchmarks, and that multiple seeds with early stopping were employed, but these details are somewhat scattered. For reproducibility, please explicitly summarize in one place the training protocol (number of seeds, early-stopping criterion, epochs, optimizer settings, batch sizes, and hardware) and ensure that the GitHub repository mirrors this exact configuration so that readers can reproduce the main tables.

7.The authors propose an effective strategy. In future work, widely developed deep learning techniques and their application to more complex scenarios (DOI: 10.1145/3664647.3681673, DOI: 10.1109/JBHI.2024.3357979) could be further explored.

**Have the authors made all data and (if applicable) computational code underlying the findings in their manuscript fully available?**

Reviewer #1: Yes

Reviewer #2: Yes

Reviewer #3: Yes

PLOS authors have the option to publish the peer review history of their article (what does this mean?). If published, this will include your full peer review and any attached files.

Reviewer #1: No

Reviewer #2: No

Reviewer #3: No

**Figure resubmission:**
---

## [Decision Letter · Decision Letter 1]

13 Jan 2026

PCOMPBIOL-D-25-02136R1

Paraplume: A fast and accurate antibody paratope prediction method provides insights into repertoire-scale binding dynamics

PLOS Computational Biology

Dear Dr. Walczak,

Thank you for submitting your manuscript to PLOS Computational Biology. After careful consideration, we feel that it has merit but does not fully meet PLOS Computational Biology's publication criteria as it currently stands. Therefore, we invite you to submit a revised version of the manuscript that addresses the points raised during the review process.

We look forward to receiving your revised manuscript.

Kind regards,

Lun Hu

Academic Editor

PLOS Computational Biology

Dominik Wodarz

Section Editor

PLOS Computational Biology

**Additional Editor Comments:**

One of our reviewers still has some concerns regarding the evaluation. Particularly, authors should seperately report the MCC and F1 scores in different cases, rather than simply averaging across all of the sequence distances.

**Reviewers' comments:**

Reviewer's Responses to Questions

**Comments to the Authors:**

Reviewer #1: The revised version is substantially better. Most of my concerns were addressed, but there are some lingering issues, one of which is a major concern on upper-bounding the performance on this dataset:

Major comments:

1. Regarding the identical arms asymmetry, the sensitivity bins of 1 to 10 and 10 to 20 are very broad. They can mask a near-linear trend in the 1 to 5 range. Also, Fig S10 suggests that a considerable number of arm pairs have perfect MCC and F1. In this case, you should report the upper bound on distance zero cases (maybe perfect MCCs and F1 scores come from there) and separately report MCC and F1 as a function of narrowly binned distances (not 1 to 10 but much smaller increments) between the two arms of the antibody. The sequence differences may be confounding MCC and F1s since you’re averaging across all of the sequence distances. This should be a straightforward reanalysis.

Minor comments:

1. Regarding the use of embeddings and the downstream tasks, I think you shouldn’t remove the negative result but instead change the language. It is important for the reader to know that your proposed weighted embedding is not a one-size-fits-all method but can yield performance improvements in some tasks. I see scientific value in all the results and in transparently reporting them. You can cite that as a limitation of the method. If you discard the negative results, the reader will mistakenly think that the paratope-weighted embeddings will work for whatever relevant downstream task they have.

2. Please specify which ESM-2 model size was used. I don't see this clearly mentioned, but there are a few different ESM-2 options with different sizes and embedding dimensions.

3. When you make scatter plots, it would be helpful not to set alpha (or opacity) to 1 so that the reader can understand the density of the dots in a given cluttered region.

4. In the beginning of page 10/29, you refer to Fig S8C, but it should be Fig S9C based on the supplementary figure tiff files attached.

Overall, the paper has made progress in terms of scoping the claims. As stated above, I have only one reservation for acceptance, which doesn't affect the presented performance results but rather the proposed upper bound on the performance.

Reviewer #2: The author made improvements to the paper, and my questions had been addressed.

Reviewer #3: The draft submitted by the author gave a satisfactory answer. I think this article is acceptable.

**Have the authors made all data and (if applicable) computational code underlying the findings in their manuscript fully available?**

Reviewer #1: Yes

Reviewer #2: Yes

Reviewer #3: None

PLOS authors have the option to publish the peer review history of their article (what does this mean?). If published, this will include your full peer review and any attached files.

Reviewer #1: No

Reviewer #2: No

Reviewer #3: No

**Figure resubmission:**
---

## [Decision Letter · Decision Letter 2]

4 Feb 2026

Dear Dr Walczak,

We are pleased to inform you that your manuscript 'Paraplume: A fast and accurate antibody paratope prediction method provides insights into repertoire-scale binding dynamics' has been provisionally accepted for publication in PLOS Computational Biology.

Best regards,

Lun Hu

Academic Editor

PLOS Computational Biology

Dominik Wodarz

Section Editor

PLOS Computational Biology

All reviewers were satisified with the changes made in this revision.

Reviewer's Responses to Questions

**Comments to the Authors:**

Reviewer #1: I appreciate the authors' work on addressing all the issues I've raised. The report looks stronger.

**Have the authors made all data and (if applicable) computational code underlying the findings in their manuscript fully available?**

Reviewer #1: Yes

PLOS authors have the option to publish the peer review history of their article (what does this mean?). If published, this will include your full peer review and any attached files.

Reviewer #1: No

---

## [Editor Report · Acceptance letter]

PCOMPBIOL-D-25-02136R2

Paraplume: A fast and accurate antibody paratope prediction method provides insights into repertoire-scale binding dynamics

Dear Dr Walczak,

I am pleased to inform you that your manuscript has been formally accepted for publication in PLOS Computational Biology. Your manuscript is now with our production department and you will be notified of the publication date in due course.

With kind regards,

Anita Estes
